# Identification of potential aggregation hotspots on Aβ42 fibrils blocked by the anti-amyloid chaperone-like BRICHOS domain

Rakesh Kumar [1], Tanguy Le Marchand[2], Laurène Adam [1], Raitis Bobrovs [3], Gefei Chen [1], Jēkabs Fridmanis[3], Nina Kronqvist [1], Henrik Biverstål [1], Kristaps Jaudzems [3], Jan Johansson [1], Guido Pintacuda [2] & Axel Abelein [1] ✉

Protein misfolding can generate toxic intermediates, which underlies several devastating diseases, such as Alzheimer's disease (AD). The surface of AD-associated amyloid-β peptide (Aβ) fibrils has been suggested to act as a catalyzer for self-replication and generation of potentially toxic species. Specifically tailored molecular chaperones, such as the BRICHOS protein domain, were shown to bind to amyloid fibrils and break this autocatalytic cycle. Here, we identify a site on the Aβ42 fibril surface, consisting of three C-terminal β-strands and particularly the solvent-exposed β-strand stretching from residues 26–28, which is efficiently sensed by a designed variant of Bri2 BRICHOS. Remarkably, while only a low amount of BRICHOS binds to Aβ42 fibrils, fibril-catalyzed nucleation processes are effectively prevented, suggesting that the identified site acts as a catalytic aggregation hotspot, which can specifically be blocked by BRICHOS. Hence, these findings provide an understanding how toxic nucleation events can be targeted by molecular chaperones.

Alzheimer's disease (AD) has been a great challenge to therapeutic management and a mechanistic understanding of the molecular processes in the disease as well as effective drugs are highly warranted[1]. Next to other prominent examples, such as Parkinson's disease and Huntington's disease, AD belongs to the family of neurodegenerative protein misfolding diseases, where AD is the most prevalent one[2]. While the mechanisms of AD-associated toxicity are still to be elucidated, the aggregation of the amyloid-β peptide (Aβ) from an unstructured form into insoluble fibrils is one hallmark of AD[3]. Recent advances have been achieved in deciphering the nucleation mechanism of Aβ in vitro, revealing the generation of new nuclei on the fibril as the dominant mechanism for Aβ40 and Aβ42 aggregation, referred to as secondary nucleation[4,5]. This nucleation reaction implies the formation of low molecular weight oligomeric or fibrillar species, which are on-pathway towards fibril formation, likely occurs upstream of

other disease-relevant processes, and is most abundant during the middle of kinetic reaction[2,6–8]. Catalytic sites on the fibril surface may hence represent "aggregation hotspots" for catalytically promoting toxic oligomer or low molecular-weight fibril assembly, which could be targeted by therapeutic agents[1]. Indeed, the mechanism of action of different anti-Aβ antibodies used in clinical trials could be linked to the specificity in preventing secondary nucleation events in in vitro kinetic assays[9]. Further, soluble fibrillar and synaptotoxic extracts from AD patients were found to share identical molecular structures as fibrils extracted from insoluble AD plaques, suggesting that they could be targeted by the same therapeutics[10]. Intriguingly, the protein domain BRICHOS, which exhibits molecular chaperone-like properties, was found to specifically prevent secondary nucleation events[11–13] and thereby attenuating secondary nucleation processes, in a superior manner than reported for the anti-Aβ antibodies[9].

[1]Department of Biosciences and Nutrition, Karolinska Institutet, 141 83 Huddinge, Sweden. [2]Université de Lyon, Centre de Resonance Magnétique Nucléaire (CRMN) à Très Hauts Champs de Lyon (UMR 5082 - CNRS, ENS Lyon, UCB Lyon 1), 69100 Villeurbanne, France. [3]Department of Physical Organic Chemistry, Latvian Institute of Organic Synthesis, Riga, Latvia. ✉e-mail: axel.abelein@ki.se

The BRICHOS domain is found in 13 protein families, where BRICHOS from Bri2 is an interesting member since it is expressed in the brain and can affect processes that are linked to AD[14]. The function of Bri2 BRICHOS apparently depends on its quaternary structure, where Bri2 BRICHOS oligomers act in the same way as classical molecular chaperones preventing amorphous aggregation of substrate proteins[12,15]. In contrast, dimers and monomers of Bri2 BRICHOS are efficient in inhibiting Aβ42 fibrillization and the monomers are particularly effective to attenuate Aβ42-associated toxic effects measured as the impact on γ-oscillations of mouse hippocampal slices[12]. Stabilized recombinant Bri2 BRICHOS monomers, using the single-point mutation R221E, target predominantly secondary nucleation processes[13]. Of importance for the development as a potential drug candidate, the monomeric form of recombinant human Bri2 BRICHOS is able to pass the blood-brain barrier in mice[16] and recent treatment studies using Aβ precursor protein (APP) knock-in mice revealed improved behavior and attenuated neuroinflammation[17], providing evidence that the in vitro findings on BRICHOS molecular mechanisms can be translated to in vivo systems[18]. These properties provide the Bri2 BRICHOS monomer with the preconditions for a therapeutic candidate as one example of a designer molecular chaperone acting against amyloid formation[14]. These chaperones are typically efficient amyloid-inhibitors already at substoichiometric ratios[12,18–20], suggesting that they can recognize specific aggregation hotspots on the fibril surface[1]. The mechanism of action of these chaperone proteins sensing specific structural properties on the fibril surface is largely unknown and to-date no molecular identification of the catalytic site targeted by a molecular chaperone is available.

In this study we employ state-of-the-art fast magic-angle spinning (MAS) solid-state nuclear magnetic resonance (NMR) with $^1H$ detection

at high magnetic field in combination with electron microscopy (EM) and other biophysical techniques to identify the interaction site of Bri2 BRICHOS monomers on Aβ42 fibrils. The insights obtained here provide a general understanding on how aggregation hotspots on Aβ42 fibrils can be blocked by molecular chaperones to prevent secondary nucleation events and generation of potentially toxic low-molecular weight species.

## Results

### Binding of Bri2 BRICHOS to Aβ42 fibrils

We set out to study the binding of the Bri2 BRICHOS to mature Aβ42 fibrils using diverse biochemical and biophysical techniques. Here, we focused on the stabilized recombinant human Bri2 BRICHOS R221E monomer mutant (subsequently referred to as BRICHOS)[13] due to the ability of Bri2 BRICHOS wildtype and R221E monomers to penetrate the blood-brain barrier[16,17] and their superior effects suppressing Aβ42-associated toxic effects compared to other Bri2 BRICHOS assembly states[12,13]. Moreover, this BRICHOS species was recently applied in intravenous treatments of AD mice models, demonstrating positive effects in behavioral studies and neuroinflammation[17].

BRICHOS is localized at the surface of Aβ42 fibrils as shown in immuno-EM images in previous reports[12,13,21]. To determine the binding affinity of BRICHOS to Aβ42 fibrils we applied surface plasmon resonance (SPR), where we applied a two-phase binding model with two association and two dissociation phases (Fig. 1A). This analysis revealed two apparent dissociation constants, one weak related to unspecific binding and one strong with a $K_D^{app}$ value of $12.9 \pm 0.2$ nM (Supplementary Table 1). This value is similar as previously obtained for binding of a monomer mutant of proSP-C BRICHOS to Aβ42 fibrils[22]. The determined value is an apparent dissociation constant,

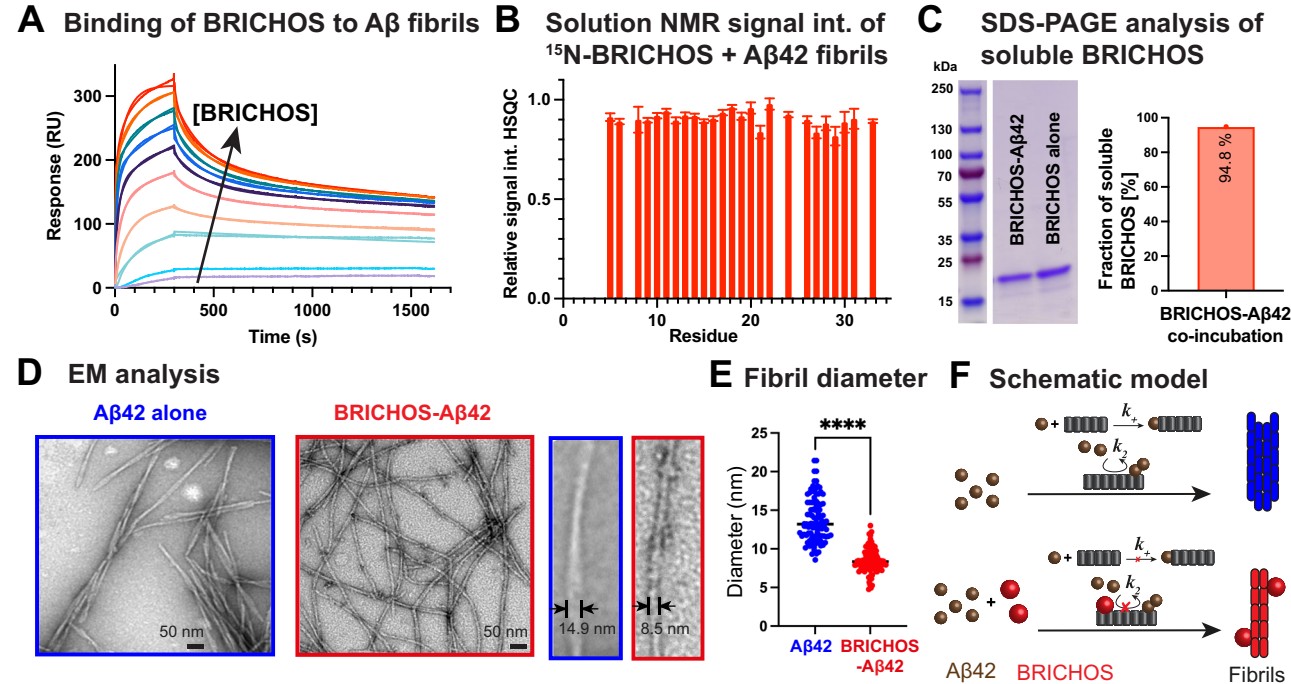

**Fig. 1 | Binding of BRICHOS to Aβ42 fibrils. A** SPR measurements revealing a dissociation constant of BRICHOS to Aβ42 fibrils of $12.9 \pm 0.2$ nM. **B** Solution NMR $^1H$-$^{15}N$ HSQC experiments of $^{15}N$-labeled BRICHOS showing an intensity decrease to $90 \pm 8$ % upon addition of Aβ42 fibrils at a 1:1 molar ratio (related to monomeric Aβ42). The error bars reflect the signal-to-noise level of one measurement (n = 1). **C** SDS-PAGE analysis of soluble BRICHOS after co-incubation with Aβ42 shows that the large proportion of BRICHOS is still soluble. The uncropped SDS-PAGE gel is shown in Supplementary Fig. 1. The experiment was repeated three times with qualitatively similar results. **D** EM images exhibiting thinner fibrils in the BRICHOS-

Aβ42 sample compared to mature Aβ42 fibrils. **E** Fibril diameter showing a reduction of a factor of around two in the presence of BRICHOS. n = 100 independent measurements are shown where the line corresponds to the mean. An unpaired two-tailed T-test was applied, where four asterisks (****) refer to $p < 0.0001$. Source data are provided as Source Data file. **F** Schematic overview about BRICHOS-modulated Aβ42 fibril formation, where BRICHOS predominantly inhibits secondary nucleation processes ($k_2$) in addition to fibril-end elongation ($k_+$) and favors the generation of thinner fibrils.

**Table 1 | Fibril diameter of Aβ42 fibrils with and without the presence of BRICHOS obtained with different fibril seeds**

| Figure 1 & Supplementary Fig. 1 | Aβ42 | mature Aβ42 fibrils + BRICHOS | BRICHOS-Aβ42 |
|---|---|---|---|
| Fibril diameter [nm] | 13.8 ± 2.9 | 15.2 ± 3.2 | 8.4 ± 1.5 |
| **Figure 2 – seeded fibrils for 3 generations** | **Aβ42 + Aβ42 seeds** | **Aβ42 + co-incubated BRICHOS-Aβ42 seeds** | **Aβ42 + BRICHOS + co-incubated BRICHOS-Aβ42 seeds** |
| Fibril diameter [nm] | 13.9 ± 2.0 | 14.0 ± 2.5 | 7.9 ± 1.4 |

Errors represent the standard derivations.

where local high concentrations of BRICHOS close to the fibril surface during the SPR measurement can decrease this value and hence overestimate the strength of binding.

The amount of BRICHOS bound to Aβ42 fibrils can be estimated using solution NMR with $^{15}$N-labeled BRICHOS, where the signal intensity of cross-peaks in $^{1}$H-$^{15}$N-HSQC experiments reports on the concentration of BRICHOS in solution. When BRICHOS binds to unlabeled Aβ42 fibrils, which are sonicated to increase the accessibility for binding, the overall tumbling time drastically increases, which is accompanied with a loss of NMR signals. Hence, the attenuation in $^{1}$H-$^{15}$N-HSQC signal intensities upon addition of Aβ42 fibrils correlates linearly with the population of bound BRICHOS. Here, we found that the signal intensity decreases uniformly for all visible cross-peaks, revealing a relative signal attenuation of ca. 10% (Fig. 1B). Of note, while chemical exchange processes can also broaden NMR signals, there is no specific increase in the line-width observable, indicating that the contribution of such processes is small. Moreover, when measuring the concentration of BRICHOS in the supernatant of a centrifuged sample containing co-incubated BRICHOS-Aβ42 fibrils, we observed that most BRICHOS is still in the soluble fraction (ca. 95%), supporting the previous results that only a small fraction, in the order of 5 to 10% of total BRICHOS is bound to Aβ42 fibrils (Fig. 1C and Supplementary Fig. 1).

### Electron microscopy image analysis of Aβ42 fibrils with BRICHOS

To study the effect of BRICHOS on the Aβ42 fibril morphology we recorded EM images of Aβ42 fibrils alone and fibrils produced by co-incubating BRICHOS with monomeric Aβ42 (Fig. 1D). We observed that a range of different fibril diameters is present for Aβ42 alone, with an average diameter of 13.8 ± 2.9 nm (Fig. 1E and Table 1). In contrast, significantly thinner fibrils are formed in the presence of BRICHOS with an average diameter of 8.4 ± 1.5 nm. Multiple measurements of the diameter revealed that co-incubated BRICHOS-Aβ42 fibrils exhibit only around half of the diameter of mature Aβ42 fibrils (Fig. 1F and Table 1). Hence, the previously described inhibitory mechanism of BRICHOS, inhibiting mainly secondary nucleation in addition to fibril-end elongation, promotes another fibril morphology (Fig. 1F). As a control, addition of BRICHOS to pre-formed Aβ42 fibrils resulted in the same fibril diameter as for mature Aβ42 fibrils (Supplementary Fig. 2 and Table 1).

To answer whether the structure of thin co-incubated BRICHOS-Aβ42 fibrils can propagate their structure, we conducted seeding experiments where we added sonicated co-incubated BRICHOS-Aβ42 fibrils as seeds to a solution of monomeric Aβ42, and performed an aggregation assay and EM analysis (Fig. 2A). The analysis of the final fibrillar state revealed that these seeded Aβ42 fibrils exhibit average diameters of 14.0 ± 2.5 nm, which indicates that the thin diameter of the seeds does not proliferate during seeding (Fig. 2B and Table 1). On the contrary, when using the same seeds in an aggregation reaction where both monomeric Aβ42 and BRICHOS were added from start, we obtained again significantly smaller average diameters of 7.9 ± 1.4 nm (Fig. 2B and Table 1). These results demonstrate that BRICHOS needs to be present during the aggregation process to produce the thin fibril morphology (Fig. 2C). While seeding generally accelerates secondary nucleation reactions, the presence of BRICHOS attenuates these secondary processes and generates a different fibril morphology (Fig. 2C).

The observed diameters correspond to molecular structures where the fibril cross-section is made by tetramers or dimers for Aβ42 alone and BRICHOS co-incubated Aβ42 fibrils, respectively. In a previous report, a mixture of dimeric and tetrameric fibril cross-sections was reported for Aβ42 fibril using mass-per-length measurements by scanning tunneling electron microscopy[23]. A later study, which applied small angle X-ray scattering (SAXS), proposed a twofold symmetric model including two filaments with two Aβ42 molecules each, where the two dimeric cross-sections are connected by the C-terminal A42 residues[24]. Based on this model, BRICHOS prevents the attachment of the two filaments, promoting a single-filament fibril (Figs. 1F and 2C).

To elucidate the basis for these effects of BRICHOS on the fibril morphology at molecular detail, we continued to prepare $^{13}$C-$^{15}$N-isotope-labeled fibrils to be investigated by MAS NMR techniques.

### Homogenous fibril preparations for MAS NMR studies

Homogenous fibril preparations are essential to achieve high-quality MAS NMR spectra suitable for structural analysis. Thus, we set out to optimize aggregation conditions to obtain homogenous Aβ42 fibrils based on reported conditions for available fibril structures[23,25,26]. Remarkably, while the core of these fibril structures agrees very well[27], the preparation protocols for the published in vitro Aβ42 fibril structures differ in several aspects: (1) the first amino acid, where some studies use an additional methionine in the N-terminus, (2) buffer conditions and additives, (3) incubation temperature, (4) quiescent vs. shaking incubation and (5) number of generations used for seeding (Supplementary Table 2). Based on these observations we chose three different conditions (Supplementary Table 3) and investigated the homogeneity of the generated fibrils by $^{13}$C-$^{13}$C DARR spectra in 3.2 mm rotors (Supplementary Fig. 3). These spectra revealed that all chosen conditions resulted in quite homogenous fibrils with only one major morphology, which is visible by only two distinct serine peaks in the DARR spectra, originating from the two serine residues in Aβ42 (Supplementary Fig. 3). Yet, using four generations with 10% parent fibrils for each generation and shaking at 25 °C gave sharper signals compared to the other conditions, and we hence selected these parameters for our further sample preparations.

### $^{1}$H-detected MAS NMR of Aβ42 fibrils with BRICHOS

To achieve proton resolution, we applied high-frequency MAS NMR using 100 kHz with ultra-small 0.7 mm rotors. The obtained $^{1}$H,$^{15}$N- and $^{1}$H,$^{13}$C-dipolar correlation spectra exhibited high quality (Fig. 3A and Supplementary Fig. 4) and, notably, the overlay with previously published spectra of Aβ42 fibrils[28] showed a very good agreement (Supplementary Fig. 4 and Supplementary Fig. 5). These results demonstrate that our Aβ42 fibrils apparently exhibit largely the same fibril structure as reported previously[23,25] and we could transfer the assignments from the literature to our spectra (Supplementary Fig. 4 and Supplementary Fig. 5). Of note, despite the rather different ways fibrils were prepared here and in literature, e.g. considering shaking/rotating vs. quiescent condition, different temperatures for aggregation, different number of seed generations and additives (Supplementary Table 2 and Supplementary Table 3), there seems to be one major dominant fibril morphology[23,25].

We subsequently prepared a sample where BRICHOS was present at a 1:1 molar ratio together with Aβ42 monomers during the aggregation

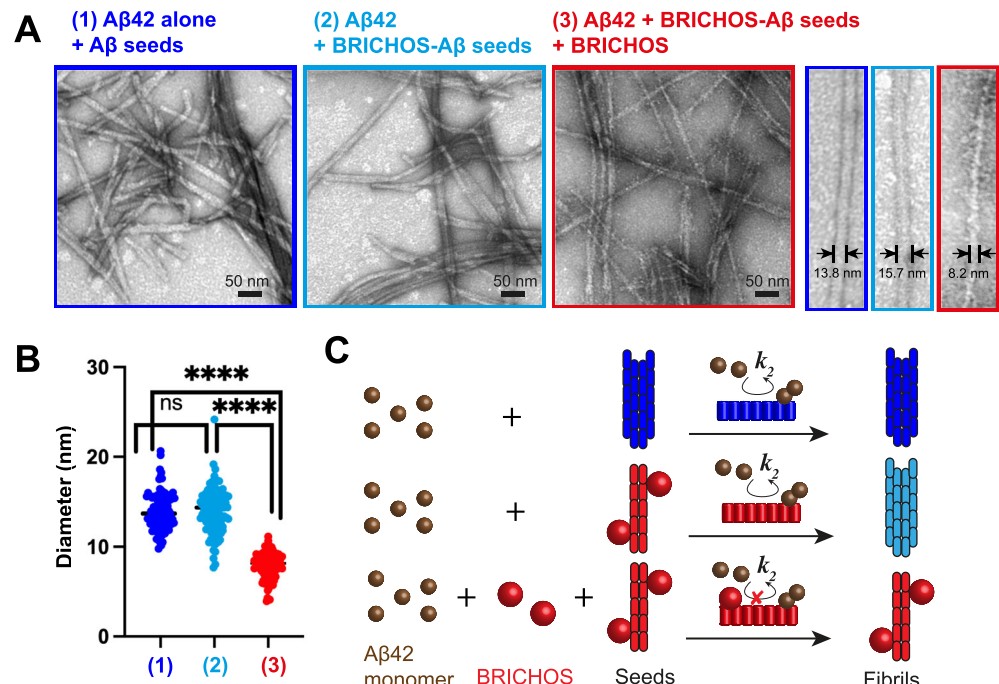

**Fig. 2 | Fibril morphology of third generation seeded fibrils. A** EM images of third generation Aβ42 fibrils prepared using (1) Aβ42 monomers & Aβ42 seeds, (2) Aβ42 monomers & BRICHOS-Aβ42 seeds and (3) Aβ42 monomers & BRICHOS & BRICHOS-Aβ42 seeds. **B** Fibril diameter of third generation seeded fibrils, exhibiting similar diameters of fibrils prepared according to (1) and (2) but half diameter fibrils for preparation (3), indicating that BRICHOS needs to be present during fibril formation to produce thinner fibrils. $n = 100$ independent measurements are shown where the line corresponds to the mean. An unpaired two-tailed T-test was applied, where four asterisks (****) refer to $p < 0.0001$ and ns refers to $p = 0.86$. Source data are provided as Source Data file. **C** Schematic overview about BRICHOS-modulated fibril formation using seeding, where mature Aβ42 fibril and BRICHOS-Aβ42 efficiently seed Aβ42 aggregation by promoting secondary nucleation processes ($k_2$), which is inhibited by the presence of BRICHOS.

reaction. The same preparation protocol was applied as for Aβ42 fibrils alone, where the final sample was obtained using four rounds of seeding where 10% of the parent fibrils were used, keeping monomeric BRICHOS and Aβ42 present at a 1:1 ratio during each seeding round.

When recording an $^1$H,$^{15}$N-correlation spectrum of the BRICHOS-Aβ42 co-incubated fibril sample, we observed a broadening of the amide $^1$H and $^{15}$N signals as compared to Aβ42 alone (Supplementary Fig. 4), indicating increased conformational disorder by co-incubation of BRICHOS (Supplementary Table 4).

The $^1$H,$^{13}$C-correlation spectrum of the BRICHOS-Aβ42 fibrils exhibits in general broader but still resolved signals and most of the cross-peaks overlay with the signals of Aβ42 alone. Yet, peak doubling was observed for several distinct signals (Fig. 3A, B), where a first set of peaks overlays with the corresponding resonances in the apo form of Aβ42 fibrils, and a second set is shifted. The shifted signals, which were subsequently assigned using 3D (H)CBCAH and (H)CCH experiments[29] (Supplementary Fig. 6), indicate a modulated local environment for certain residues, which is caused by the presence of BRICHOS. Highlighting the residues exhibiting peak doubling on the Aβ42 fibril structure (Fig. 3C) revealed that all affected residues are close and located in the three C-terminal β-strands. Interestingly, the perturbed residues include the two continuous hydrophobic patches along the surface of the fibril core constituting the dimer-dimer interface, as highlighted from SAXS measurements of Aβ42 fibrils[24] (Fig. 3D).

Finally, to investigate whether the observed chemical shift perturbations (Fig. 3A, B) can also be produced by BRICHOS bound to mature Aβ42 fibrils, we prepared a second sample where BRICHOS was added to mature fibrils at 1:1 molar ratio (referred to the initial Aβ monomer concentration). This sample again exhibits $^1$H,$^{15}$N- and $^1$H,$^{13}$C-correlation spectra that largely overlay with the spectra of the Aβ fibril alone sample (Supplementary Fig. 4, Supplementary Fig. 7 and

Supplementary Fig. 8). Interestingly, while the effects are much less pronounced, also this sample shows chemical shift doubling, with the shifted peaks displaying the same trend as for the co-incubated BRICHOS-Aβ42 sample, although weaker intensity (Supplementary Fig. 7 and Supplementary Fig. 8). Due to the tetrameric conformation of Aβ42 mature fibrils the theoretical binding surface is decreased by a factor two. Other experimental conditions, such as fibril bundling, further diminish the available binding surface, making the second set of shifted signals less visible.

## Solvent accessibility of fibrils revealed by PRE experiments

To have further insight on the effect of BRICHOS binding, we measured solvent paramagnetic relaxation enhancement (PRE) experiments on both Aβ42 and BRICHOS-Aβ42 fibrils. These measurements allow probing the solvent accessibility of residues by comparing site-specific relaxation rates (here $^{15}$N $R_1$) of the target molecule in the presence and in the absence of a solubilized paramagnetic dopant (here, 100 mM CuEDTA). Residues exposed to the solvent exhibit enhanced relaxation rates (PREs), while buried residues are unaffected by the presence of the paramagnetic center. Without the presence of BRICHOS, we observed significant PREs for seven residues (Fig. 4A, blue bars, and Supplementary Fig. 9), all located in solvent accessible segments (Fig. 4B). The broad $^1$H,$^{15}$N-correlation peaks for the BRICHOS-Aβ42 co-incubated sample (Supplementary Fig. 4) makes a PRE analysis difficult and we hence measured PRE values on the sample where BRICHOS was added to mature fibrils. We recorded significantly lower PRE values for residues 26–29 (Fig. 4). This region largely coincides with the solvent-exposed site exhibiting chemical shift changes (Fig. 3). BRICHOS binding may thus bury parts of residues 26-29 on the fibril surface, causing the decreased PRE values due to reduced solvent accessibility. Interestingly, the C-terminal residue A42 shows higher PRE values in the samples with BRICHOS, which could be explained by

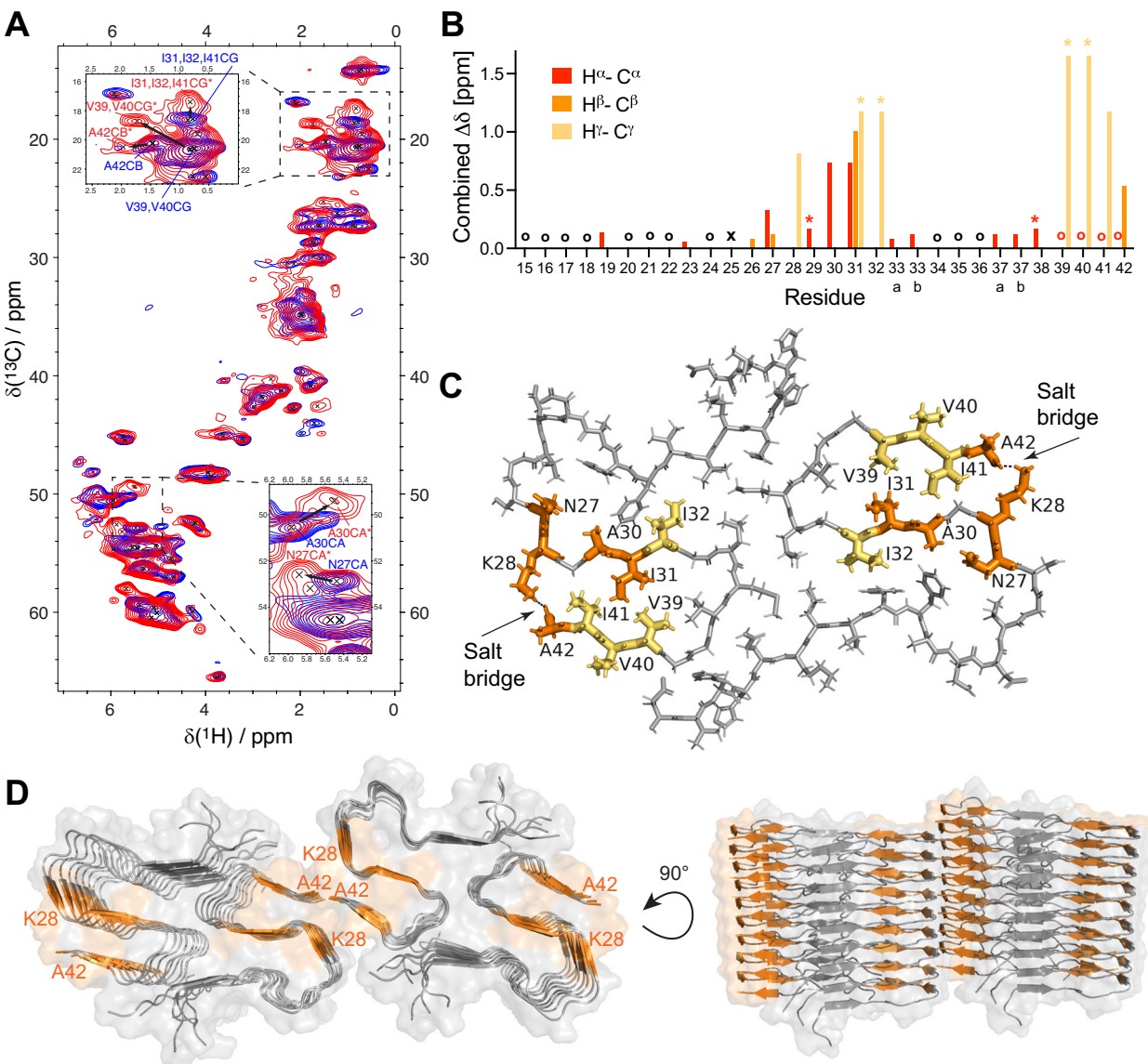

**Fig. 3 | Chemical shift changes of BRICHOS-Aβ42 fibrils and structural model.**
**A** ¹H,¹³C-correlation spectra of Aβ42 fibrils alone (blue) and BRICHOS-Aβ42 co-incubated fibrils (red). The doubled cross-peaks in the BRICHOS-Aβ42 spectrum are labeled in red with a star (*). The insets represent different zoomed regions.
**B** Combined (¹H and ¹³C) chemical shift changes between the new signals and those observable in the spectra of Aβ42 fibrils alone. Circles refer to overlap in the spectrum, stars to ambiguous assignments and crosses to residues with missing assignment. **C** Residues exhibiting signals with significant chemical shift doubling are colored in orange and yellow (for ambiguous assignments) on the fibril structure[23], revealing that the last three β-strands, including the salt bridge between K28 and A42, are affected by the presence of BRICHOS. **D** Such residues are also illustrated onto the 3D model of tetrameric Aβ42 fibrils[24].

a structural change of the C-terminus upon BRICHOS binding, making the C-terminus more solvent-accessible.

## Modeling BRICHOS–Aβ42 fibril binding

Based on the identified chemical shift perturbations and PRE constraints, the interaction of Aβ42 fibrils with BRICHOS can be modeled using the HADDOCK protein docking software[30,31]. The modeling was constrained such that the residues of BRICHOS predicted to interact with its natural amyloidogenic client, the Bri23 peptide, are the active residues, which are located in face A[14] (Supplementary Fig. 10). Prior to the docking, the AlphaFold2 model of BRICHOS was relaxed using molecular dynamics (MD) simulation, resulting in a relocation of α-helix 1, which partly exposed face A. Additionally, for Aβ42 the missing N-terminal residues in the fibril structure were added computationally, and relaxed using MD. The resulting complex of the HADDOCK docking revealed an ionic interaction and hydrogen bonding of BRICHOS

(in particular D139, D141 and/or D158) with residue K28 on the Aβ42 fibril (Fig. 5 and Supplementary Fig. 10). Importantly, while the docking calculations revealed slightly different positions of BRICHOS on the fibril surface, multiple top-scoring complexes contained this ionic interaction network, indicating the essential role of such contacts in the interaction (Supplementary Fig. 10). Notably, the side chain of K28 forms a salt bridge with A42 C-terminal carboxyl group in the Aβ fibril structure, which can be perturbed by the presence of BRICHOS and allows a structural reorientation of the C-terminus.

## Discussion

Kinetic insights into the processes of Aβ42 self-assembly have pointed out the generation of new nucleation units on the fibril surface as the dominant mechanism for self-replication of fibril mass[4], which is also the major source for generation of potentially neurotoxic oligomeric or low-molecular-weight fibrillar Aβ42 species[8]. Specific inhibition of

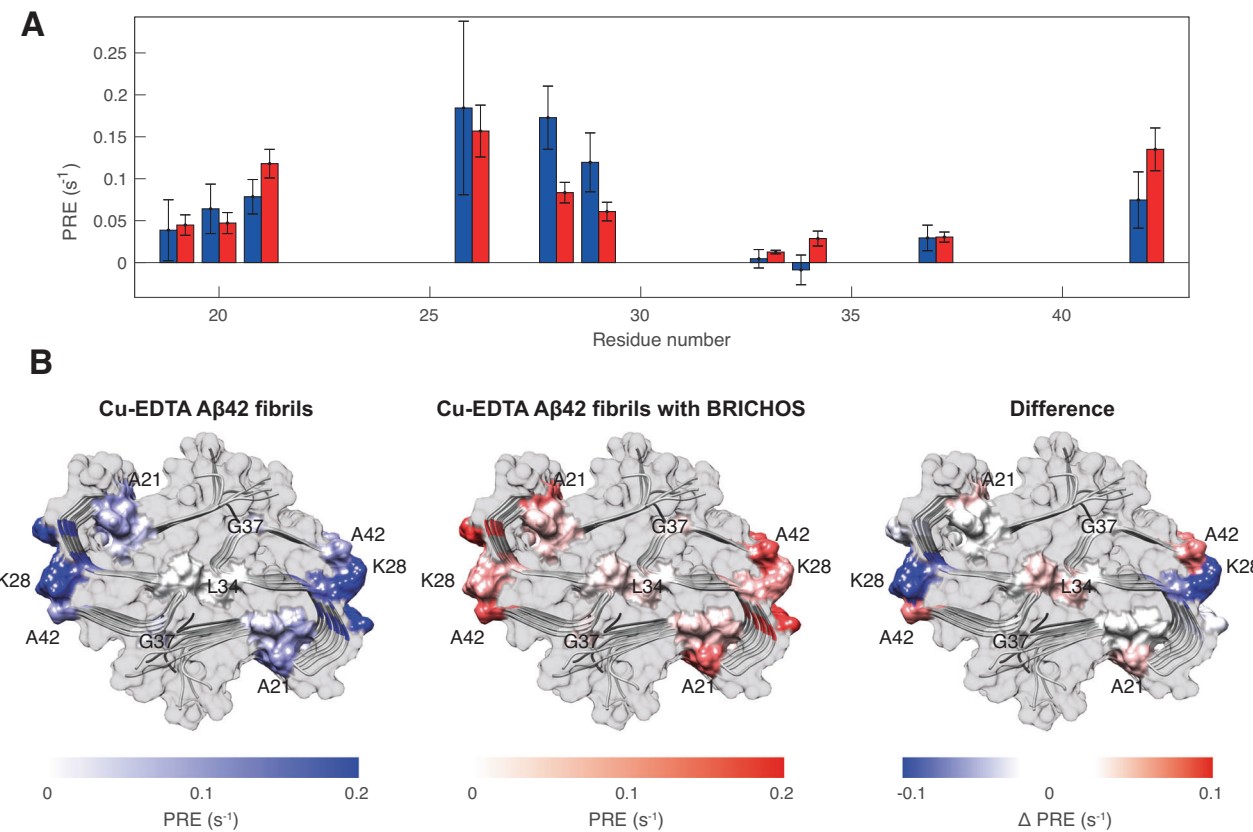

**Fig. 4 | PRE experiments revealing solvent accessible sites. A** Solvent [15]N long-itudinal PRE values upon addition of Cu-EDTA to Aβ42 fibrils alone (blue bars) and in the presence of BRICHOS (red bars). Error bars were estimated from the experimental noise by use of a Monte-Carlo evaluation of one measurement ($n = 1$). **B** PRE values are colored onto the dimeric structure of Aβ42 fibrils[23], respectively in blue for Aβ42 fibrils alone (left), in red for Aβ42 fibrils with BRICHOS (middle), and as a blue-to-red gradient (right), where the blue parts correspond to residues more exposed in Aβ42 fibrils alone and red parts correspond to residues more exposed in Aβ42 fibrils with BRICHOS.

this process by molecular chaperone-like proteins has been shown to result in greatly attenuated toxic effects[11–13,32]. Despite that their binding affinity to Aβ42 fibrils is not very high and only a small fraction is bound to the fibrils, molecular chaperones can efficiently inhibit secondary nucleation process[11–13,32], suggesting that there are specific aggregation hotspots on the fibril surface, which can be blocked by designer molecular chaperones[1]. In the context of a recent treatment study showing that intravenously injected BRICHOS can reduce the amyloid plaque amount and neuroinflammation as well as improve cognitive behavior in APP-knock in mouse models[17], suppression of Aβ42 oligomer generation in vitro can seemingly be translated to treatment effects in vivo[18].

Here, we report molecular insights into the structural modulation of the Aβ42 fibril surface by BRICHOS, providing a molecular picture of the mechanism on how molecular chaperones and chaperone-like proteins can inhibit Aβ42 self-assembly by binding to the fibril surface. We found that the structure of the three C-terminal β-strands is affected by the presence of BRICHOS, representing a potential binding site of BRI-CHOS (Fig. 5). Further, co-incubation of BRICHOS promotes the formation of Aβ42 fibrils, which exhibit only half of the fibril diameter compared to Aβ42 alone fibrils (Fig. 1D–F). Interestingly, this filament morphology cannot be proliferated in seeding assays except if BRICHOS is present during the aggregation reaction (Fig. 2), indicating that the single filament morphology is generated by steric hindrance of the build-up of two filaments by BRICHOS. The measurement of solvent accessibility through PRE effects suggested that BRICHOS does not only promote different fibril morphology but indeed transiently binds to β-strand region of residues 26-29 (Fig. 4). Modeling of the binding using HADDOCK pinpoints toward a dominant interaction of BRICHOS with K28 (Fig. 5 and Supplementary Fig. 10). Due to the salt bridge between K28 and A42 in the mature Aβ42 fibril structure, BRICHOS binding presumably destabilizes the C-terminal part of the fibril structure, allowing an alternative, more solvent exposed conformation. Indeed, the increased PRE value of A42 supports this scenario (Fig. 4).

Our results suggest that the three C-terminal β-strands, and in particular the solvent-exposed β-strand containing residues 26-28, play a key role in catalyzing secondary nucleation events on the fibril surface and hence constitute a suitable target site for molecular chaperones or designed drugs. Notably, the cross-β structure is found as a common feature in all typical amyloid fibrils, providing a plausible explanation of the generic ability of BRICHOS to bind diverse amyloid fibril structures[21] and inhibit secondary nucleation process on these fibrils, as reported for the isoform Aβ40[33] and the familial arctic E22G Aβ42 variant[34]. Worth to mention is that the current Aβ42 NMR structures do not include the largely unstructured, fuzzy N-terminal region[23,25,26], and our MAS NMR measurements are hence blind for potential additional interactions of BRICHOS with the N-terminal part of Aβ42.

Based on these and previous findings, we propose the molecular mechanisms of action of inhibiting secondary nucleation processes by BRICHOS on the Aβ42 fibrils surface (Fig. 5). While the overall fibril structure is largely conserved, BRICHOS specifically modulates the C-terminal β-strands, creating a solvent-exposed binding site for BRI-CHOS, in particular represented by the β-strand built up by residues 26-28. Of note, the current results identify the interacting site in two dimensions, yet the present approach is blind towards localization of bound BRICHOS along the fibril axis. The binding site may hence also be responsible for catalyzing the generation of new nucleation units, representing an aggregation hotspot on the fibril surface. Targeting

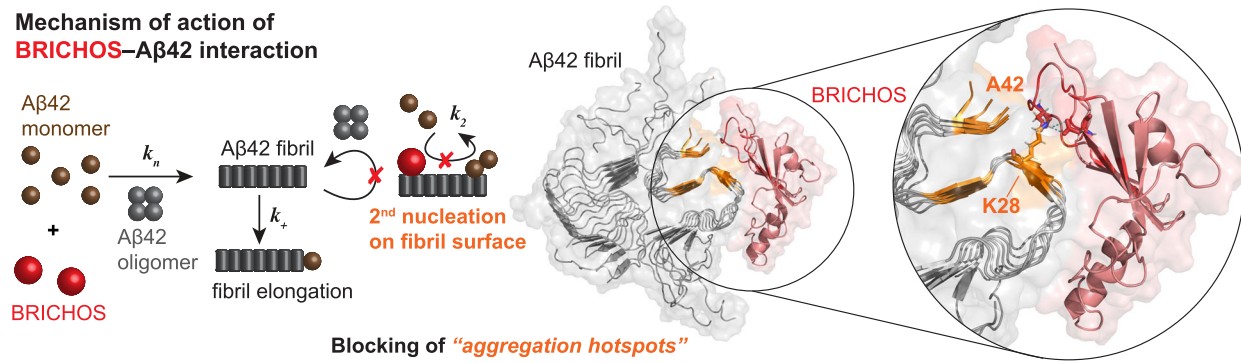

**Fig. 5 | Mechanistic model of BRICHOS binding to Aβ42 fibril and inhibition of secondary nucleation by blocking "aggregation hotspots" on the fibril surface.** The scheme shows a 3D model of a chemical-shift driven docking of BRICHOS (in red) onto the dimeric fibril structure[23], where the N-terminal residues were added computationally. The simulated BRICHOS-Aβ42 complex reveals an ionic network interaction involving K28, which disturbs the salt bridge in the Aβ42 fibril between K28 and A42. Hence, the C-terminal β-strands and in particular the solvent exposed β-strand, stretching between residues 26–28, may represent the "aggregation hotspot" for secondary nucleation, which can be blocked by BRICHOS.

this catalytic site should thus facilitate the attenuation of toxic effects associated with Aβ42 secondary nucleation reactions, creating a stepping-stone for the design of future AD therapeutics.

## Methods

### Peptide and protein production

Monomeric Aβ42 was produced using a spider silk-derived solubility tag[35,36]. In brief, pT7 plasmid containing a TEV recognition site with His6-NT*$_{FISp}$-Aβ$_{42}$ gene was expressed in BL21 (DE3) cells in 1 l LB medium at 30 °C with agitation of 110 rotation per minute. Expression was induced by adding 0.5 mM Isopropyl β-D-1-thiogalactopyranoside (IPTG) when the cell density (OD$_{600nm}$) reached a value of 0.8–0.9. Further, the temperature of the incubator was reduced to 20 °C and the cells were grown overnight. $^{15}$N- and $^{13}$C-labeled Aβ42 was expressed in same manner except that the culture medium was M9 instead of LB where $^{15}$NH$_4$Cl (1 g/l) and $^{13}$C-glucose (2 g/l) was used. The cells were pelleted the next day at 7000 × g for 20 min at 4 °C and then resuspended in 40 ml 20 mM Tris-HCl, pH 8.0. 20 ml of resuspended cells with 8 M urea were sonicated (2 s on, 2 s off, 65 % power) for 5 min with a probe sonicator. The cell debris were pelleted by centrifugation at 22,000 g for 30 min at 4 °C. The supernatant was loaded on a 20 ml Ni-NTA column (GE Healthcare) and then washed with 15 mM urea in using 20 mM Tris-HCl, pH 8 and 8 M urea. The bound protein was eluted with 300 mM imidazole in 20 mM Tris-HCl, pH 8 and 8 M urea. The protein was dialyzed overnight at 4 °C to remove urea and then it was cleaved overnight with TEV protease (1:30 enzyme to substrate, w/w) in 20 mM Tris-HCl pH 8, 0.5 mM EDTA and 1 mM DTT. The cleaved protein was lyophilized overnight and the next day it was dissolved in 10 ml of 7 M guanidine-HCl. The protein was loaded on a Superdex 30 26/600PG column after equilibration with 20 mM sodium phosphate, pH 8.0. Aβ42 peptide containing fractions were collected and pooled. The Aβ42 purity was checked by running a SDS PAGE gel.

A similar solubility tag was applied to purify R221E Bri2 BRICHOS monomers[13]. R221E BRICHOS protein was expressed in SHuffle T7 competent *Escherichia coli* cells. The cells were grown at 30 °C in LB medium and protein expression was induced by adding 0.5 mM Isopropyl β-D-1-thiogalactopyranoside when OD$_{600 nm}$ of cells reaches ~0.8. After induction, cells were grown at 20 °C overnight. The cells were pelleted at 7000 × g for 20 min at 4 °C and resuspended in 20 mM Tris-HCl pH 8. The cells were sonicated on ice for 5 minutes (2 s on, 2 s off, 65% power) and then pelleted by centrifugation at 22,000 g for 30 min at 4 °C. The fusion protein was cleaved overnight by thrombin (1:1000 enzyme to substrate, w/w) at 4 °C. It was purified further by loading the protein on Ni-NTA column where the His$_6$-NT*-tag binds to the column whereas BRICHOS elutes in the flowthrough. The flowthrough was finally purified by size exclusion chromatography using a Superdex 75 PG column and ÄKTA system (GE Healthcare, UK). The peak containing monomeric BRICHOS was collected and pooled and its purity was checked by SDS PAGE. The protein was either used the same day or stored at −80 °C for further use.

### Sample preparation for MAS NMR

All fibrils were obtained by aggregation of monomeric Aβ42, performed at room temperature with agitation in 20 mM sodium phosphate buffer (pH 8, 0.02% NaN3, 0.2 mM EDTA), where three rounds of seeding were performed to obtain homogenous fibrils. Aβ42 fibril alone sample was prepared from 30 µM Aβ42 monomers under agitation at 200 rpm at 37 °C for 24 h. The fibrils were sonicated to produce seeds (2 s pulse on/off for 3 min at 20% amplitude). For each seeding generation 10% seeds of the previous generation were used. Finally, 30 µM $^{13}$C-$^{15}$N-labeled Aβ42 were incubated in 50 ml falcon tube at 37 °C for 3 days adding 10% seeds of the third generation and using a shaker at 200 rpm. The $^{13}$C-$^{15}$N-labeled fibrils were collected by pelleting down the fibrils at 17,000 g for 60 min in 1.5 ml microfuge tube.

BRICHOS-Aβ42 co-incubated fibrils were obtained by co-incubating 30 µM Aβ42 monomers in the presence of equimolar Bri2 BRICHOS R221E monomer under agitation 200 rpm at 37 °C for 24 h. Similarly as described for Aβ alone fibrils, three generations of seeds were produced where at each step BRICHOS was present at 1:1 molar ratio compared to Aβ42 monomers.

Seeding experiments for EM analysis were performed using 10% of parent fibrils for three generation as described above. For seeding experiments with BRICHOS-Aβ42 co-incubated fibrils, BRICHOS was either present only for the parent fibril generation or during all fibril generations.

For the PRE experiments, the rotor containing $^{15}$N-labeled fibrils was opened and soaked for 1 h into a buffer solution containing 100 mM CuEDTA in an ultracentrifuge at 100,000 g and 10 °C.

### EM measurements

5 µl of Aβ42 fibrils was drop casted on formvar/carbon-coated 400 mesh copper grids and washed twice with 5 µl MQ water after incubation on grid for 10 min. Samples were stained with 5 µl of 1% uranyl formate and excess stained was wiped with Whatman filter paper after 3 min. Samples were allowed to air dry at room temperature. Imaging was performed using a FEI Tecnai 12 Spirit BioTWIN microscope, operated at 100 kV with 2 x 2 k Veleta CCD camera (Olympus Soft Imaging Solutions, GmbH, Münster, Germany). For each sample, 10-15 images were obtained randomly at magnification between 20,000x–60,000x. ImageJ (1.52 K, Wayne Rasband, NIH, USA) software was used for analysis of

fibril diameters, where 100 different fibrils were analyzed. Statistical tests were performed using an unpaired two-tailed T-test calculated with GraphPad Prism 9, where four asterisks (****) refer to $p < 0.0001$.

## MAS NMR measurements

MAS NMR spectra for optimization of Aβ42 aggregation conditions were acquired on an 800 MHz Bruker Avance III HD spectrometer equipped with a 3.2 mm $^{13}C/^{15}N(^{1}H)$ E-free MAS probe and run by TopSpin 3.6.5. The sample temperature was regulated with variable-temperature gas flow set to 273 K. The fibril homogeneity was assessed by comparison of $^{13}C$-$^{13}C$ DARR spectra[37] acquired at a MAS frequency of 17 kHz. The same spectra acquisition and processing parameters were used for all three samples: cross-polarization (CP) from $^{1}H$ to $^{13}C$ using a linear ramp from 60 to 48 kHz on $^{1}H$, 65 kHz on $^{13}C$ and a contact time of 1.2 ms, Spinal64 decoupling[38] at 83 kHz field amplitude during acquisition. The number of scans was 32, acquisition time was 10 ms and the recycle delay was 2.5 s giving a total experiment time of about 18 h. Spectra were processed by applying a 60° shifted squared sine-bell function in both dimensions.

The MAS NMR experiments for assignment and comparison of the chemical shifts were performed on a Bruker Neo 18.8 T spectrometer ($^{1}H$ frequency of 800 MHz), run by TopSpin 4.1.4 using a 3-channel (HCN) 0.7 mm MAS probe. Sample rotation frequency was maintained at $100.00 \pm 0.02$ kHz, and the sample temperature was approximately 15 °C. The $^{1}H,^{15}N$ and $^{1}H,^{13}C$ correlations were acquired by following, with little modifications, the sequence introduced in Refs. [39,40]. Cross polarization between $^{1}H$ and the heteronuclei was performed with 1 ms and 0.5 ms contact times for the forward and backwards transfers, respectively. A 10% upward linear ramp was applied on the $^{1}H$ channel, with RF powers optimized around 170 kHz for the $^{1}H$ channel and 70 kHz for $^{15}N$ and $^{13}C$. Offsets for $^{15}N$ and $^{13}C$ channels were set to 117.5 ppm and 40.0, while $^{1}H$ offset was set on resonance with $H_2O$ line (approx. 4.8 ppm).

Assignments of $^{1}H,^{15}N$ and $^{1}H,^{13}C$ spectra for Aβ42 alone were based on published chemical shift lists[28]. Shifted cross-peaks in the co-incubated BRICHOS-Aβ42 were obtained by analysis of two additional $^{1}H$-detected 3D experiments, namely an Hα-detected (H)CBCAHA experiment and a (H)CCH TOCSY, described in Ref. [29]. The (H)CBCAHA and (H)CCH TOCSY experiments were based on the above described (H)CH experiment, with additional $^{13}CB$ to $^{13}CA$ INEPT transfer step (7.1 ms echo delay) or additional $^{13}C$-$^{13}C$ mixing (WALTZ-16, during 11.5 ms), respectively.

The combined chemical shift change was calculated as

$$\left| \delta_{comb} \right| = \sqrt{\left( \frac{\delta_C}{2} \right)^2 + \delta_H^2}. \tag{1}$$

For the sample containing Aβ42 fibrils alone, relaxation experiments were measured at 23.5 T (1000 MHz $^{1}H$ Larmor frequency) based on a $^{1}H,^{15}N$ $^{1}H$-detected CP-HSQC experiment incorporating an inversion recovery relaxation delay ranging from 0 to 10 s. The acquisition software was Topspin 4.0.3. For the mature fibrils in presence of BRICHOS, these experiments were measured at 18.8 T (800 MHz $^{1}H$ Larmor frequency) with inversion recovery delays ranging from 0 to 20 s. The relaxation rates were obtained by fitting the experimental decay curves with a mono-exponential function. The error was estimated from the experimental noise by use of a Monte-Carlo evaluation, with 500 simulations.

All NMR spectra were processed using Topspin 4.0 and NMRpipe[41] and spectra were analyzed by NMRFAM-Sparky[42].

## Modeling of BRICHOS-Aβ42 fibril complex

Docking models of BRICHOS to Aβ42 fibrils were created using the HADDOCK v2.4 webserver[30,31] with the basic parameter set. The Aβ42 fibril model suitable for docking studies was prepared based on the

PDB structure 5KK3 [https://doi.org/10.2210/pdb5KK3/pdb]. The 10 missing N-terminal amino acids were added to the Aβ42 fibril structure, and 100 ns long molecular dynamics simulation was conducted to relax the system and obtain an ensemble of full-length Aβ42 fibril structures suitable for docking. Molecular dynamic simulation was performed in Desmond using OPLS4 force field. Five frames were randomly selected from the last 10 ns of the simulation and were used in the docking studies. The BRICHOS structure used in docking studies was based on an AlphaFold2 model[43]. The full length Bri2 model was truncated to retain only the structured residues 110-266 and the R221E mutation was introduced to match the system used in the experiments. The BRICHOS system suitable for molecular docking was created in a similar way as Aβ42 fibril – five R221E Bri2 BRICHOS conformations were randomly selected from the last 10 ns of 100 ns long molecular dynamics simulation.

The residues within 5 Å from the amyloidogenic part of Bri2, referred to as Bri23, were chosen as interacting residues for the BRICHOS (residues 115, 130, 132, 139-143, 145 and 156-158). For Aβ42, residues exhibiting significant chemical shift changes were selected as the interacting residues (residues 27, 28, 30-32 and 40-42). The Aβ42 interaction site was constrained to three monomeric subunits in the middle of the fibril.

Each generated BRICHOS conformation was docked to each Aβ42 conformation, producing complexes with Haddock scores in range from −96.6 to −145.9 (see Supplementary Fig. 10). The top-scoring complexes were visually examined for contacts involving specified interacting residues. MD models and the best HADDOCK complexes are provided as Supplementary Data 1.

## SPR measurements

SPR analysis was performed on a BIAcore 3000 instrument (BIAcore AB). Aβ42 fibrils were prepared by sonication[22] and immobilized by amine coupling onto one flow-cell on a C1 sensor chip (GE Healthcare) using 20 mM sodium phosphate, 0.2 mM EDTA, pH 8.0 as running buffer and a flow rate of 20 μL/min⁻¹. A blank reference surface was prepared using the same coupling protocol without injection of fibrils. The flow-cells were stabilized in HBS-E (10 mM HEPES, 150 mM NaCl, 0.2 mM EDTA, pH 7.5) running buffer overnight. R221E Bri2 BRICHOS diluted in HBS-E to 9 different concentrations ranging from 0.195 – 50 μM were injected in duplicates at a flow rate of 30 μL/min⁻¹ and the surfaces were regenerated between each sample by a 30 s injection of 20 mM NaOH and washing with running buffer. The response from the immobilized surface was subtracted with the response from the blank surface for each injected concentration.

SPR sensorgrams were fitted with a two-phase association and two-phase dissociation model given by:

$$k_{obs1} = c \cdot k_{on1} + k_{off2} \tag{2}$$

$$k_{obs2} = c \cdot k_{on2} + k_{off1} \tag{3}$$

$$R_{Asso}(t) = k_{on1} \cdot c \cdot \frac{B_{max1}}{k_{on1} \cdot c + k_{off2}} \cdot \left(1 - \exp\left(-k_{obs1} \cdot t\right)\right) \\ + k_{on2} \cdot c \cdot \frac{B_{max2}}{k_{on2} \cdot c + k_{off1}} \cdot \left(1 - \exp\left(-k_{obs2} \cdot t\right)\right) \tag{4}$$

$$R_{Diss}(t) = R_0 \cdot \left( a \cdot \exp\left(-k_{off1} \cdot (t - t_0)\right) \\ + (1 - a) \cdot \exp\left(-k_{off2} \cdot (t - t_0)\right) \right) \tag{5}$$

where $k_{on}$ and $k_{off}$ refer to the on- and off-rate constants, $k_{obs1}$ and $k_{obs2}$ are the observed association rate constants, $c$ is the BRICHOS concentration, $B_{max1}$ & $B_{max2}$ and $R_0$ are the amplitudes for association

and dissociation, respectively, and *a* is a value between 0 and 1, describing the contribution of the two dissociation phases. The association and dissociation phases were fitted simultaneously where the lowest two BRICHOS concentrations were excluded from the global fit analysis.

## Solution NMR with $^{15}$N labeled BRICHOS in the presence of sonicated Aβ42 fibrils

Sonicated Aβ42 sonicated fibrils were prepared from 133 µM Aβ42 monomers under agitation at 200 rpm at 37 °C for 72 h. The samples were then centrifuged at 20 000 g for 10 min, the supernatant was removed, and the remaining pellet was resuspended in 20 mM sodium phosphate buffer, 0.2 mM EDTA, pH 7.4. The fibrils were sonicated (2 s pulse on/off for 3 min at 20 % amplitude) and $^{1}$H-$^{15}$N HSQC spectra were recorded using 74 µM of R221E Bri2 BRICHOS monomers, 10% D$_2$O using a 700 MHz Bruker spectrometer with a cryogenic probe at 25 °C. The sonicated Aβ42 fibrils were added to the BRICHOS solution at equimolar concentration (ratio 1:1) and the same experiment was repeated. Error bars were derived from the signal-to-noise ratio.

## Reporting summary

Further information on research design is available in the Nature Portfolio Reporting Summary linked to this article.

## Data availability

The authors declare that the data supporting the findings of this study are available within the paper and its Supplementary Information files. MD models and the best HADDOCK complexes are provided as Supplementary Data 1. Should any raw data files be needed in another format they are available from the corresponding author upon reasonable request. The Aβ42 fibril model was prepared based on the PDB structure 5KK3 [https://doi.org/10.2210/pdb5KK3/pdb]. Source data are provided with this paper.

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

## Acknowledgements

We are thankful for financial support from FORMAS (A.A.), Swedish Society for Medical Research (A.A.), Alzheimer foundation (A.A.), Åke Wiberg foundation (A.A.), Magnus Bergvall foundation (A.A., H.B.), Åhlen foundation (A.A.), KI Research Foundation Grants (A.A.), Foundation for Geriatric Diseases KI (A.A.), Swedish Research Council (J.J.), Swedish Brain Foundation (J.J.) and JPco-fuND/EU PETABC 2020-02905/EC 643417 (H.B.). This work benefited from access to RALF-NMR and has been supported by iNEXT-Discovery, project number 15812, funded by the Horizon 2020 program of the European Commission. We additionally acknowledge financial support from IR INFRANALYTICS FR2054 for enabling this research. We thank Zhiyu (Jeff) Sun (CRMN Lyon) for fruitful discussions on the interpretation of the NMR data.

## Author contributions

R.K., T.L.M., L.A., R.B., G.C., J.F., N.K. & A.A. performed experiments; R.K., T.L.M., G.C., L.A., J.F., N.K., H.B., K.J., J.J., G.P. & A.A. analyzed data and results; J.J., G.P. & A.A. supervised study; A.A. together with R.K., T.L.M., J.J., G.P. wrote the article.

## Funding

## Competing interests

The authors declare no competing interests.
