## [Peer Review File · Nature Communications]

Identification of potential aggregation hotspots on A β 42 fibrils blocked by the anti-amyloid chaperone-like BRICHOS domainREVIEWER COMMENTS

Reviewer #1 (Remarks to the Author):

Kumar et al. report to have identified hotspots on the surface of amyloid-beta where secondary nucleation takes place, based on the interaction with Brichos. Brichos is a domain with chaperone function that was previously shown to efficiently inhibit secondary nucleation. The authors use a range of structural methods (SPR, EM, solution and solid-state NMR) to characterise the interaction between amyloid-beta fibrils and Brichos. Guided by NMR chemical shift changes, they dock Brichos onto a previously established structure of amyloid-beta fibrils to arrive at their model, which they validate using paramagnetic relaxation enhancement. While it is of great interest to elucidate the binding mode of Brichos, as well as the sites where secondary nucleation takes place, the current study is not convincing as detailed below.

Major comments:

1. In the title and abstract, the authors suggest that they have identified the hotspots on the surface of amyloid-beta fibrils where secondary nucleation takes place, by mapping the interaction with Brichos at these sites. However, the interface that they show is rather large and the details of the interaction are not described, nor do they inform on how or where on this surface amyloid-beta monomers would interact to undergo secondary nucleation. So the data do not support the claim that aggregation hotspots have been identified.
2. The authors make a point of the fact that only a small amount of Brichos binds to the fibrils. However, the 1:1 ratio they use is a large excess given the fact that one Brichos molecule will cover multiple amyloid-beta monomers on the fibril surface, so it is not surprising that the majority of Brichos stays in solution. In fact, the structural models show one Brichos binding across ~10 amyloid-beta monomers judged from Figures 3 and 4, leaving 90% of Brichos sterically unable to bind. Moreover, I would consider the nM affinity for the specific binding event from the SPR measurements strong, not weak, binding. So effectively the entire fibril surface is covered with Brichos under these circumstances, which makes it unsurprising that secondary nucleation would be blocked. Again, this means that the authors cannot claim to have identified hotspots.
3. The authors do not provide sufficient evidence that their preparations are in agreement with the structures that they use to interpret the data. The ssNMR spectra of the fibrils alone overlay reasonably well with the previously published assignments, but not perfectly. Moreover, the lines become very broad when Brichos is included. Since it was present from the start of the aggregation reaction, it cannot be excluded that the resulting morphology differs from the one without Brichos. In fact, some of the

residues that are affected are not exposed in the fibril structure (A30-I32) and it seems unlikely that they are able to interact directly with Brichos. Shifts in CA and CB as seen for these residues are actually more indicative of a structural rearrangement than a surface interactions. The analysis would be much easier if Brichos were added to preformed fibrils, which is arguably also more physiological than including it in a 1:1 ratio from the beginning.

4. Related to the point above, the assignments of the shifted peaks were done with an (H)CBCAH experiment that connects atoms within the residue. Additional experiments connecting to neighbouring residues would be required to provide more confidence of the assignments.

5. The problem with docking is that you get out what you put in. The authors predetermined the interacting residues of Brichos based on the interactions with its native substrate, but they would need to show that these are also involved in the interaction with amyloid-beta fibrils. Furthermore, the parts of the Brichos domain that would block this interface were removed, further steering the docking. In the amyloid-beta fibril structure, the flexible N-termini are moreover missing. These should be added computationally before performing the docking procedure, since they may occlude certain parts of the fibril. Finally, the authors would need to show more details on the binding interface and present a comparison with alternative docking solutions to demonstrate that their result is the most likely mode of interaction.

6. The PRE data are inconsistent with the structural models of the tetrameric fibrils and the dimeric fibrils with Brichos bound (figure 4). The region comprising A21 is not involved in the interface between the dimers, so it is not expected to become more solvent exposed when Brichos is bound. For A42, without Brichos half of the residues would be buried in the interface between the dimers and half would be exposed, whereas binding of Brichos should shield all of them. So here also it is not expected that the accessibility increases. The parts shown in blue become less accessible even though this is not where Brichos binds according to the model. Finally there is no real argument why the result for L34 would be an artefact.

7. Brichos inhibits amyloid-beta aggregation very efficiently, so it is somewhat surprising that adding it in a 1:1 ratio still leads to fibril formation. A ThT assay should be done to demonstrate that the end point of the reaction is reached, because otherwise one might be comparing a final state with an intermediate one. Again adding Brichos to preformed fibrils would also solve this issue.

8. The authors refer to toxic oligomers or toxic nucleation events in several instances. However, the existence/role of toxic oligomers is heavily debated. The authors mention 'increasing lines of evidence' citing two reviews from 2012 and 2014. Since then, the key paper demonstrating toxic oligomers in vivo turned out to be fraudulent, and recently it was shown that the "soluble oligomers" from brain extracts are actually amyloid-beta fibrils (Stern et al, Neuron 2023). Moreover, neuronal cell death and cognitive

decline correlate with tau aggregation, which occurs downstream of amyloid-beta, which also goes against a role for toxic amyloid-beta intermediates (e.g. Ossenkoppele et al, Brain 2016). Altogether I would suggest the authors to phrase this more carefully. Secondary nucleation is the main process that accelerates amyloid-beta aggregation, which ultimately plays a role in disease, so that in itself is sufficient motivation to try to understand this process.

Minor comments:

1. In Figure S1, the molecular weights of the marker bands should be included.
2. In figure S3, it would be helpful to draw a box around the serine peaks that are mentioned in the text.
3. The legends of figure S4B and figure S6 should explain what the different colors are.
4. In the strips in figure S6, the scale of the ^1H frequency is missing.

Reviewer #2 (Remarks to the Author):

Kumar et al. " Identification of aggregation hotspots on A β 42 fibrils blocked by an anti-amyloid chaperone-like domain"

This manuscript describes a solid-state NMR and TEM study of the impact of the chaperone Brichos on the Ab42 fibril morphology and structure. By observing chemical shift changes and fibril widths as a function of the method of Brichos addition to the A β 42 fibril, the authors conclude that the chaperone binds the C-terminal b-strands of the peptide and interferes with secondary nucleation of the fibrils. As a result they conclude that the C-terminal region of the peptide is the aggregation hotspot of A β 42.

The data shown in this work are of good quality, and the interpretation may be correct. However, some important data analysis are inadequately presented, and various structural model figures are poorly made. So until these are revamped it's difficult to know if the conclusions are all correct.

Figure 2A: please show zoomed-in regions of the TEM images, with a scale bar much smaller than 500 nm. The scale bar is preferably at 100 nm, to allow visual inspection of the fibril width change.

Figure 3A and Figure S4B: given the line broadening caused by Brichos, how did the authors obtain the precise chemical shift perturbation values? Please show zoomed-in areas of the residues that are shifted and please also provide a chemical shift perturbation bar diagram. Indicate in this diagram which residues cannot be measured.

Please clarify what is the ratio of bound Brichos to the number of Ab monomers in the fibril. Based on the solution NMR data and SDS-PAGE gel, do you think it is 1 : 10 Brichos to Ab or 1:20? And is the measured stoichiometry consistent with the docked model of Brichos to Ab? What's the length of Brichos along the fibril? Presumably it is smaller than the length of 10 b-strands (which would be ~48 Å) or even 20 b-strands. Can you explain the low binding ratio of Brichos to Ab? Since the authors have a sample of ¹⁵N-labeled Brichos with unlabeled Ab, can you measure a simple 2D hNH spectrum of Ab bound Brichos to obtain some qualitative information about the bound chaperone's conformational dynamics?

Figure 1F and Figure 2C: the schematic models of how Ab42 fibril elongation and secondary nucleation are inhibited by Brichos are hard to understand and color mismatched. Please indicate in brown color where the Ab42 monomers would be in the elongated fibril in both panels. Presumably the brown monomers would be on the end of the blue fibril when Brichos is absent. In the middle row where the existing dimeric fibril has Brichos on the side, are there supposed to be brown Ab42 monomers at the end of the cyan tetrameric fibrils?

Figure 3B, C: the structure rendering of Brichos on Ab42 fibrils in this paper is very difficult to see and no details can be gleaned. Please add arrows to indicate 90° rotation between different panels of Fig. 3B and 3C. Please lighten the grey color for Ab42 and show the protein ribbons and heavy atoms more clearly. If Brichos is in the grey mass in Figure 3B, then please use a different color for Brichos. In Figure 3C: Please choose a different shade than grey to indicate Brichos. Then add a figure panel to show a zoomed-in view of the Ab42 residues (in sticks) that are perturbed by Brichos.

Please also add a figure to indicate the active residues in Brichos that are modeled in Haddock, and in the same figure show the Ab residues that interact with Brichos as seen from the chemical shift perturbation.

Reviewer #3 (Remarks to the Author):

The search for means to combat protein assembly diseases was immeasurably advanced with the discovery of a secondary nucleation pathway, which had the two-fold effects of accelerating the process and accounting for the bulk of the fibril mass. The present work takes this mechanism to a new level by the identification of structural origins of the secondary nucleation process, thanks to the adherence of BRICHOS, a domain found in 13 protein families with chaperone activity. Though BRICHOS suppression of secondary nucleation has previously been demonstrated, here solid-state magic angle spinning NMR demonstrates that affected residues are close and located in the three C-terminal beta strands. Model building and paramagnetic resonance enhancement experiments supported these findings. This is thus a well documented and very exciting finding that can have an impact in developing therapeutics as well as probing the fundamental features underlying this assembly.

The only caveat here is that fibrils were created by agitation, and there remains a remote chance that the agitation favors a pathway that, while real, might not be the central one. Similarly, the electron microscopy entails deposition onto grids that again is likely to disrupt features that are not collinear (e.g. the double fibrils observed). Something like an AFM study on non-agitated solutions could thus be welcome in the future. But this thirst for more data should not be taken as undercutting this elegant piece of work, presented in admirable clarity.

REVIEWER COMMENTS

Reviewer #1 (Remarks to the Author):

Kumar et al. report to have identified hotspots on the surface of amyloid-beta where secondary nucleation takes place, based on the interaction with Brichos. Brichos is a domain with chaperone function that was previously shown to efficiently inhibit secondary nucleation. The authors use a range of structural methods (SPR, EM, solution and solid-state NMR) to characterise the interaction between amyloid-beta fibrils and Brichos. Guided by NMR chemical shift changes, they dock Brichos onto a previously established structure of amyloid-beta fibrils to arrive at their model, which they validate using paramagnetic relaxation enhancement. While it is of great interest to elucidate the binding mode of Brichos, as well as the sites where secondary nucleation takes place, the current study is not convincing as detailed below.

We thank the reviewer for pointing out the interest of this study and have addressed the concerns as described below.

Major comments:

1. In the title and abstract, the authors suggest that they have identified the hotspots on the surface of amyloid-beta fibrils where secondary nucleation takes place, by mapping the interaction with Brichos at these sites. However, the interface that they show is rather large and the details of the interaction are not described, nor do they inform on how or where on this surface amyloid-beta monomers would interact to undergo secondary nucleation. So the data do not support the claim that aggregation hotspots have been identified.

We agree with the reviewer that the current study does not reveal any details about how A β monomers are attached to the fibril surface to undergo secondary nucleation.

However, the current study does provide molecular insights on *where* the process of secondary nucleation takes place on the fibril surface, using BRICHOS as a tool to identify the catalytic site for secondary nucleation. Our results determine the BRICHOS binding site to a well-defined region on the fibril surface, where residues forming the three C-terminal β -strands are affected. Interestingly, the new data added in our revised manuscript, using a sample where BRICHOS was added to mature fibrils resulting in updated PRE analysis and docking simulations (as suggested by this reviewer below), emphasize the importance of the solvent-exposed β -strand between residues 26-28. The current findings hence delineate a well-defined region, which is targeted by BRICHOS, and that consequently is the site where secondary nucleation occurs. Notably, this site is well defined in two dimensions, while the current experiments do not allow specification of the BRICHOS binding density along the fibrils, see further below.

We have now updated our manuscript and clarified these points.

2. The authors make a point of the fact that only a small amount of Brichos binds to the fibrils. However, the 1:1 ratio they use is a large excess given the fact that one Brichos molecule will cover multiple amyloid-beta monomers on the fibril surface, so it is not surprising that the majority of Brichos stays in solution. In fact, the structural models show one Brichos binding across \sim 10 amyloid-beta monomers judged from Figures 3 and 4, leaving 90% of Brichos sterically unable to bind. Moreover, I would consider the nM affinity for the specific binding event from the SPR measurements strong, not weak, binding. So effectively the entire fibril surface is covered with Brichos under these circumstances, which makes it unsurprising that secondary nucleation would be blocked. Again, this means that the authors cannot claim to have identified hotspots.

BRICHOS bound to the A β surface covers ca. six A β molecule layers and due to the fibril symmetry BRICHOS binding is possible from two sites. Importantly, the binding is

localized to a specific site, in particular the β -strand between residue 26-28, leaving large parts of the fibrils surface uncovered. Particularly, we have clarified that “aggregation hotspots” refer to a catalytic site, which is well defined in two dimensions although the current data lack information on BRICHOS position along the fibril axis.

The SPR data represent an apparent dissociation constant, which might overestimate the actual strength of the binding due to high local concentrations of BRICHOS close to the fibril surface. Importantly, Bri2 BRICHOS used in this study is only able to delay A β aggregation to some degree (Chen et al. Nat Commun, 2017 and Chen et al., Commun Biol, 2019), which makes it unlikely that the whole fibril surface would be blocked by BRICHOS so that no secondary nucleation could occur.

We have pointed out and clarified these limitations in the updated manuscript.

3. The authors do not provide sufficient evidence that their preparations are in agreement with the structures that they use to interpret the data. The ssNMR spectra of the fibrils alone overlay reasonably well with the previously published assignments, but not perfectly. Moreover, the lines become very broad when Brichos is included. Since it was present from the start of the aggregation reaction, it cannot be excluded that the resulting morphology differs from the one without Brichos. In fact, some of the residues that are affected are not exposed in the fibril structure (A30-I32) and it seems unlikely that they are able to interact directly with Brichos. Shifts in CA and CB as seen for these residues are actually more indicative of a structural rearrangement than a surface interactions. The analysis would be much easier if Brichos were added to preformed fibrils, which is arguably also more physiological than including it in a 1:1 ratio from the beginning.

We thank the reviewer for this suggestion and performed new experiments where we added BRICHOS to mature A β fibrils. We found that the perturbation of the spectrum is overall much less pronounced compared to the co-incubated BRICHOS-A β sample, but, interestingly, several peaks show the same trends and are doubled with the same pattern as in the sample where BRICHOS was added to preformed fibrils. Hence, we conclude that the observed chemical shift perturbations are caused by binding of BRICHOS, leading to a modulated structural arrangement of the C-terminus.

We updated the manuscript with these new results and include the new analysis and data in the updated Figure 3 and SI Figure S7, see below.

Figure 3. Chemical shift changes of BRICHOS-A β 42 fibrils and structural model. (A) ^1H , ^{13}C -correlation spectra of A β 42 fibrils alone (blue) and BRICHOS-A β 42 co-incubated fibrils (red). The doubled cross-peaks in the BRICHOS-A β 42 spectrum are labeled in red. The insets represent different zoomed regions. **(B)** Combined chemical shift changes are plotted, where circles refer to overlap in the spectrum, stars to ambiguous assignments and crosses to residues with missing assignment. **(C)** Residues exhibiting signals with significant chemical shift differences are colored in orange and yellow (for ambiguous assignments) on the fibril structure²³, revealing that the last three β -strands, including the salt bridge between K28 and A42, are affected by the presence of BRICHOS. **(D)** Such residues are also illustrated onto the 3D model of tetrameric A β 42 fibrils²⁴.

Fig. S7. $^1\text{H},^{13}\text{C}$ -correlation spectra. The $^1\text{H},^{13}\text{C}$ -correlation spectra of A β 42 fibrils alone (blue), BRICHOS added to mature A β 42 fibrils (green) and BRICHOS-A β 42 co-incubated samples (red) are shown. The inserts represent different zoomed regions. The assignment of the shifted peaks of the BRICHOS-A β 42 co-incubated sample is shown in red.

4. Related to the point above, the assignments of the shifted peaks were done with an (H)CBCAH experiment that connects atoms within the residue. Additional experiments connecting to neighbouring residues would be required to provide more confidence of the assignments.

The assignment was performed using hCBCAH and hCCH-TOCSY experiments, which we found sufficient to assign the few peaks that exhibit chemical shift changes and the limited number of peaks visible for A β 42. There were however some ambiguities in the assignment (e.g. overlap of CG of V39/V40). Therefore, we have updated the graph (Figure 3C) showing the chemical shift changes to distinguish between strong assignments and ambiguous assignments, see updated Figure 3 above.

5. The problem with docking is that you get out what you put in. The authors predetermined the interacting residues of Brichos based on the interactions with its native substrate, but they would need to show that these are also involved in the interaction with amyloid-beta fibrils. Furthermore, the parts of the Brichos domain that would block this interface were removed, further steering the docking. In the amyloid-beta fibril structure, the flexible N-termini are moreover missing. These should be added computationally before performing the docking procedure, since they may occlude certain parts of the

fibril. Finally, the authors would need to show more details on the binding interface and present a comparison with alternative docking solutions to demonstrate that their result is the most likely mode of interaction.

We thank the reviewer for these suggestions and have performed new MD simulations and docking calculations.

First, we performed a MD simulation of the AlphaFold2 structure of BRICHOS, which resulted in a relocation of α -helix 1, exposing parts of face A of BRICHOS. Of note, in the updated calculations, we subsequently did not remove any parts of BRICHOS.

While we currently do not have any experimental data pinpointing which are the interacting residues of BRICHOS, previous experimental results suggest that face A is involved in fibril interactions (Willander et al., PNAS, 2011 and Leppert et al., Protein Sci, 2023). Moreover, the AlphaFold2 model of full-length Bri2 including the Bri23 region, which is the natural amyloidogenic client peptide, demonstrates that Bri23 is bound to face A. Due to the lack of other experimental data, we hence assigned the interacting residues of BRICHOS based on the hypothesis that the same residues bind to the A β 42 fibril surface. Prior to the HADDOCK docking, we added computationally the missing N-terminal part to the A β fibril and performed an MD simulation to relax the structure before docking.

The obtained best docking models revealed a predominant interaction with K28 on the fibrils surface, which affects the salt bridge between K28 and A42. Thus, the new MD and docking calculations provide additional support for the identity of the binding surface.

We included these results in the updated Figure 5 and SI Figure S10, see below, as well as the corresponding parts in updated manuscript.

Figure 5. Mechanistic model of BRICHOS binding to A β 42 fibril and inhibition of secondary nucleation by blocking "aggregation hotspots" on the fibril surface. The scheme shows a 3D model of a chemical-shift driven docking of BRICHOS (in red) onto the dimeric fibril structure²³, where the N-terminal residues were added computationally. The simulated BRICHOS-A β 42 complex reveals an ionic network interaction involving K28, which disturbs the salt bridge in the A β 42 fibril between K28 and A42. Hence, the C-terminal β -strands and in particular the solvent exposed β -strand, stretching between residues 26-28, may represent the "aggregation hotspot" for secondary nucleation, which can be blocked by BRICHOS.

Fig. S10. HADDOCK models of BRICHOS-A β 42 complexes. (A) AlphaFold2 model of R221E Bri2 BRICHOS aligned with the Alpha2 model of full-length Bri2 protein where the natural amyloidogenic client, the Bri23 peptide, is shown. Residues close (within 5 Å) to Bri23 peptide are marked in red. **(B)** Haddock score of modeled complexes is presented. **(C)** The three best Haddock models are visualized where the interacting residues on A β 42 fibril are marked in orange and on BRICHOS in red. The zoom-in graph shows the ionic interaction network between K28 on the A β 42 fibril surface and D139, D141 and D158 of BRICHOS.

6. The PRE data are inconsistent with the structural models of the tetrameric fibrils and the dimeric fibrils with Brichos bound (figure 4). The region comprising A21 is not involved in the interface between the dimers, so it is not expected to become more solvent exposed when Brichos is bound. For A42, without Brichos half of the residues would be buried in the interface between the dimers and half would be exposed, whereas binding of Brichos should shield all of them. So here also it is not expected that the accessibility increases. The parts shown in blue become less accessible even though this

is not where Brichos binds according to the model. Finally there is no real argument why the result for L34 would be an artefact.

We have now performed PRE experiments on the new sample where BRICHOS was added to mature fibrils. This data set is of much higher quality and exhibits narrower peak widths. The availability of 2D ^1H , ^{15}N -fingerprint correlations allowed us to notably record PRE data for a larger pool of amide groups, with a higher quality than that permitted on the BRICHOS-A β 42 co-incubated sample. In the previous sample, the analysis was confounded by the double occurrence of BRICHOS binding and change in fibril architecture (dimer vs. tetramer) and, additionally, broad lines may have led to an artifactually overestimated solvent accessibility for some sites (e.g. L34). The scenario is simplified for the new sample, and the new dataset confirms the decreased solvent accessibility of residues 26-29 while does not show an increased solvent accessibility of residues 30-32. Moreover, our updated MD simulations, see also our detailed answer to point (5) above, provide additional evidence that BRICHOS preferably binds to the β -strand containing K28, disturbing the salt bridge of K28 to A42, which may be the explanation for the observed for the increased PRE value for A42, which can be assigned to a structural rearrangement of the C-terminus.

We have now updated our manuscript and prepared a new Figure 4, see below.

Figure 4. PRE experiments revealing solvent accessible sites. (A) Solvent ^{15}N longitudinal PRE values upon addition of Cu-EDTA to A β 42 fibrils alone (blue bars) and in the presence of BRICHOS (red bars). (B) PRE values are colored onto the dimeric structure of A β 42 fibrils²³, respectively in blue for A β 42 fibrils alone (left), in red for A β 42 fibrils with BRICHOS (middle), and as a blue-to-red gradient (right), where the blue parts correspond to residues more exposed in A β 42 fibrils alone and red in A β 42 fibrils with BRICHOS.

7. Brichos inhibits amyloid-beta aggregation very efficiently, so it is somewhat surprising that adding it in a 1:1 ratio still leads to fibril formation. A ThT assay should be done to demonstrate that the end point of the reaction is reached, because otherwise one might be comparing a final state with an intermediate one. Again adding Brichos to preformed fibrils would also solve this issue.

While BRICHOS is an excellent tool to specifically inhibit secondary nucleation, the overall bulk aggregation is only delayed at 1:1 molar ratio and eventually reaches the

same final ThT intensity as without BRICHOS, which has been previously published (Chen et al., *Commun Biol* **3**, 32, 2020), see Figure below.

Figure 3A from Chen et al., *Commun Biol* (2020): Individual fits (solid lines) of normalized and averaged aggregation traces (crosses) of 3 μ M A β 42 in the presence of 0 (black), 10 (purple), 30 (cyan), 50 (yellow), 70 (green), and 100% (blue) rh Bri2 BRICHOS R221E monomer.

Moreover, we have measured the ThT intensity of small aliquots taken from samples incubated in the same manner as used for the ssNMR samples (and repeated this experiment twice). These measurements demonstrate that the final ThT intensities are saturated at the end of the aggregation reaction. These results are now included in SI Figure S3, see below.

Taken together, these results show that both samples represent the final state of the aggregation reaction. Finally, we have performed the suggested additional ssNMR experiments, where BRICHOS was added to mature fibrils, see reply to point (3) above.

Fig. S3. ¹³C-¹³C DARR spectra, EM images of different sample preparations for Aβ₄₂ alone fibrils and final ThT signal intensity. (A) ¹³C-¹³C DARR spectra and (B) EM images of different sample preparations as described in SI Table S2. Sample 3 exhibits the sharpest signals and the two distinct peaks at around 55-60 ppm correspond to the two serine signals (marked in the box), indicating one major fibril morphology. (C) Final ThT intensities of Aβ₄₂ alone and BRICHOS-Aβ₄₂ co-incubated samples at different time points. The samples were incubated in the same manner as used for the preparation of the ssNMR samples, where small aliquots were taken and the ThT intensity was measured in a ThT plate using four replicates. The individual errors represent the standard deviations of these four replicates and the experiment was repeated twice, referred to as (a) and (b).

8. The authors refer to toxic oligomers or toxic nucleation events in several instances. However, the existence/role of toxic oligomers is heavily debated. The authors mention 'increasing lines of evidence' citing two reviews from 2012 and 2014. Since then, the key paper demonstrating toxic oligomers in vivo turned out to be fraudulent, and recently it was shown that the "soluble oligomers" from brain extracts are actually amyloid-beta fibrils (Stern et al, Neuron 2023). Moreover, neuronal cell death and cognitive decline correlate with tau aggregation, which occurs downstream of amyloid-beta, which also goes against a role for toxic amyloid-beta intermediates (e.g. Ossenkoppele et al, Brain 2016). Altogether I would suggest the authors to phrase this more carefully. Secondary

nucleation is the main process that accelerates amyloid-beta aggregation, which ultimately plays a role in disease, so that in itself is sufficient motivation to try to understand this process.

We thank the reviewer for the suggested literature and discussion. We have now updated the introduction and discussion part, incorporated the reviewer's suggestions and added the literature references.

Minor comments:

1. In Figure S1, the molecular weights of the marker bands should be included. We apologize for the missing labels and have updated the SI Fig. S1.

Fig. S1. SDS-PAGE analysis of BRICHOS binding to A β 42 fibrils. Different concentrations of BRICHOS (1.5 to 10 μ M) were co-incubated with different concentrations of A β 42 (3, 6 and 10 μ M) to form fibrils. The samples were centrifuged and the supernatant loaded on the SDS-PAGE. The results show that most BRICHOS is still in the supernatant and not bound to the fibrils.

2. In figure S3, it would be helpful to draw a box around the serine peaks that are mentioned in the text.

We thank for this suggestion and updated the figure accordingly, please see SI Figure S3 shown above.

3. The legends of figure S4B and figure S6 should explain what the different colors are.

We apologize for the missing explanation and updated the manuscript with consistent color coding, also including the new sample of BRICHOS added to mature fibrils.

Fig. S4. ^1H , ^{15}N -correlation spectra of A β 42 fibrils and the effect of BRICHOS. (A) ^1H , ^{15}N -correlation spectra of A β 42 fibrils alone (blue) overlaid with the spectrum and assignment as published by Griffin & coworkers. (pink) ¹. **(B,C)** Overlap of ^1H , ^{15}N -correlation spectra of A β 42 fibrils alone (blue) with (B) BRICHOS-A β 42 co-incubated fibrils (red) and (C) BRICHOS added to mature A β 42 fibrils (green).

Fig. S6. Assignment of shifted peaks of BRICHOS-A β 42 using (H)CBCAH experiment. Spectra are shown for A β 42 fibrils alone (blue) and BRICHOS-A β 42 fibrils (red). Strips of (H)CBCAH experiment of the three shifted peaks in the CA-region using previous assignment of A β 42 alone fibrils¹.

4. In the strips in figure S6, the scale of the 1H frequency is missing. We apologize for the unclear labeling and have made the scale for the 1H frequency better visible, please see SI Figure S6 above.

Reviewer #2 (Remarks to the Author):

Kumar et al. " Identification of aggregation hotspots on A β 42 fibrils blocked by an anti-amyloid chaperone-like domain"

This manuscript describes a solid-state NMR and TEM study of the impact of the chaperone Brichos on the A β 42 fibril morphology and structure. By observing chemical shift changes and fibril widths as a function of the method of Brichos addition to the A β 42 fibril, the authors conclude that the chaperone binds the C-terminal b-strands of the peptide and interferes with secondary nucleation of the fibrils. As a result they conclude that the C-terminal region of the peptide is the aggregation hotspot of A β 42.

The data shown in this work are of good quality, and the interpretation may be correct. However, some important data analysis are inadequately presented, and various structural model figures are poorly made. So until these are revamped it's difficult to know if the conclusions are all correct.

Figure 2A: please show zoomed-in regions of the TEM images, with a scale bar much smaller than 500 nm. The scale bar is preferably at 100 nm, to allow visual inspection of the fibril width change.

We thank the reviewer for this suggestion and updated Figures 1 & 2 with better scale bars and zoomed-in images.

Figure 1. Binding of BRICHOS to A β 42 fibrils. (A) SPR measurements revealing a dissociation constant of BRICHOS to A β 42 fibrils of 12.9 ± 0.2 nM. (B) Solution NMR ^1H - ^{15}N HSQC experiments of ^{15}N -labeled BRICHOS showing an intensity decrease to 90 ± 8 % upon addition of A β 42 fibrils at a 1:1 molar ratio (related to monomeric A β 42). (C) SDS-PAGE analysis of soluble BRICHOS after co-incubation with A β 42 shows that the large proportion of BRICHOS is still soluble. The uncropped SDS-PAGE gel is shown in SI Figure S1. (D) EM images exhibiting thinner fibrils in the BRICHOS-A β 42 sample compared to mature A β 42 fibrils. (E) Fibril diameter showing a reduction of a factor of around two in the presence of BRICHOS. (F) Schematic overview about BRICHOS-modulated A β 42 fibril formation, where BRICHOS predominately inhibits secondary nucleation processes (k_2) in addition to fibril-end elongation (k_+) and favors the generation of thinner fibrils.

Figure 2. Fibril morphology of third generation seeded fibrils. (A) EM images of third generation A β 42 fibrils prepared using (1) A β 42 monomers & A β 42 seeds, (2) A β 42 monomers & BRICHOS-A β 42 seeds and (3) A β 42 monomers & BRICHOS & BRICHOS-A β 42 seeds. **(B)** Fibril diameter of third generation seeded fibrils, exhibiting similar diameters of fibrils prepared according to (1) and (2) but half diameter fibrils for preparation (3), indicating that BRICHOS needs to be present during fibril formation to produce thinner fibrils. **(C)** Schematic overview about BRICHOS-modulated fibril formation using seeding, where mature A β 42 fibril and BRICHOS-A β 42 efficiently seed A β 42 aggregation by promoting secondary nucleation processes (k_2), which is inhibited by the presence of BRICHOS.

Figure 3A and Figure S4B: given the line broadening caused by Brichos, how did the authors obtain the precise chemical shift perturbation values? Please show zoomed-in areas of the residues that are shifted and please also provide a chemical shift perturbation bar diagram. Indicate in this diagram which residues cannot be measured. We have now updated Figure 3 including zoom in areas of the affected residues and provide chemical shift perturbation bar diagrams included in the updated Figure 3, see below. In this figure we plot the chemical shift perturbations of all nuclei that can clearly be assigned.

Figure 3. Chemical shift changes of BRICHOS-A β 42 fibrils and structural model. (A) ^1H , ^{13}C -correlation spectra of A β 42 fibrils alone (blue) and BRICHOS-A β 42 co-incubated fibrils (red). The doubled cross-peaks in the BRICHOS-A β 42 spectrum are labeled in red. The insets represent different zoomed regions. **(B)** Combined chemical shift changes are plotted, where circles refer to overlap in the spectrum, stars to ambiguous assignments and crosses to residues with missing assignment. **(C)** Residues exhibiting signals with significant chemical shift differences are colored in orange and yellow (for ambiguous assignments) on the fibril structure²³, revealing that the last three β -strands, including the salt bridge between K28 and A42, are affected by the presence of BRICHOS. **(D)** Such residues are also illustrated onto the 3D model of tetrameric A β 42 fibrils²⁴.

Please clarify what is the ratio of bound Brichos to the number of Ab monomers in the fibril. Based on the solution NMR data and SDS-PAGE gel, do you think it is 1 : 10 Brichos to Ab or 1:20? And is the measured stoichiometry consistent with the docked model of Brichos to Ab? What's the length of Brichos along the fibril? Presumably it is smaller than the length of 10 β -strands (which would be ~ 48 Å) or even 20 β -strands. Can you explain the low binding ratio of Brichos to Ab? Since the authors have a sample of 15N-labeled Brichos with unlabeled Ab, can you measure a simple 2D hNH spectrum of Ab bound Brichos to obtain some qualitative information about the bound chaperone's conformational dynamics?

The usage of different methods allowed us to confirm and determine a range for the binding of BRICHOS to A β , which is between 1:10 and 1:20. The actual value is dependent of the current experimental set-up. BRICHOS covers approximately six β -strand of different molecule layers. We have clarified that in the revised manuscript. Please see also our reply to reviewer 1 point (2), above.

As suggested, we also investigated a sample where ^{15}N -labeled A β 42 fibrils were co-incubated with ^{13}C -labeled BRICHOS. Only a very weak ^{13}C NMR signal was detected for this preparation using a dipolar-based coherence transfer scheme (Figure below). This sensitivity penalty prevents any further site-specific characterization of the bound chaperone. When comparing the intensity of the ^{13}C envelope to that of the ^{15}N signals of ^{15}N -labeled A β 42, we note however that the intensity is significantly lower than that expected from a 1:10 binding stoichiometry. This result points towards the existence of substantial μs -ms dynamics affecting the conformation and/or the position of BRICHOS when bound to fibrils.

Figure: (H)NH (red) and (H)CH (blue) 1D traces of a sample containing ^{15}N -labelled Abeta mature fibrils and ^{13}C -labelled BRICHOS. The intensities are normalized relatively to the signal accumulation (16 and 448 repetitions for (H)NH and (H)CH, respectively). A 28 times magnification of the (H)CH signal is shown in the inset.

Figure 1F and Figure 2C: the schematic models of how Ab42 fibril elongation and secondary nucleation are inhibited by Brichos are hard to understand and color mismatched. Please indicate in brown color where the Ab42 monomers would be in the elongated fibril in both panels. Presumably the brown monomers would be on the end of the blue fibril when Brichos is absent. In the middle row where the existing dimeric fibril has Brichos on the side, are there supposed to be brown Ab42 monomers at the end of the cyan tetrameric fibrils?

We have now updated Figure 1 and 2, where we explicitly show the different nucleation steps and how they are modulated by BRICHOS, please see Figures above. Unfortunately, our current data based on TEM and previous aggregation kinetics results do not allow making conclusions about the location of the monomers in the fibrils.

Figure 3B, C: the structure rendering of Brichos on Ab42 fibrils in this paper is very difficult to see and no details can be gleaned. Please add arrows to indicate 90° rotation between different panels of Fig. 3B and 3C. Please lighten the grey color for A β 42 and show the protein ribbons and heavy atoms more clearly. If Brichos is in the grey mass in Figure 3B, then please use a different color for Brichos. In Figure 3C: Please choose a different shade than grey to indicate Brichos. Then add a figure panel to show a zoomed-in view of the Ab42 residues (in sticks) that are perturbed by Brichos.

We have now updated Figure 3 and moved the BRICHOS-A β 42 complex to Figure 5. Please see Figures 3 above and Figure 5 below.

**Mechanism of action of
BRICHOS-A β 42 interaction**

Figure 5. Mechanistic model of BRICHOS binding to A β 42 fibril and inhibition of secondary nucleation by blocking "aggregation hotspots" on the fibril surface. The scheme shows a 3D model of a chemical-shift driven docking of BRICHOS (in red) onto the dimeric fibril structure²³, where the N-terminal residues were added computationally. The simulated BRICHOS-A β 42 complex reveals an ionic network interaction involving K28, which disturbs the salt bridge in the A β 42 fibril between K28 and A42. Hence, the C-terminal β -strands and in particular the solvent exposed β -strand, stretching between residues 26-28, may represent the "aggregation hotspot" for secondary nucleation, which can be blocked by BRICHOS.

Please also add a figure to indicate the active residues in Brichos that are modeled in Haddock, and in the same figure show the Ab residues that interact with Brichos as seen from the chemical shift perturbation.

We thank the reviewer for this suggestion and prepared a new SI Figure S10, see below.

A Bri2 BRICHOS**B HADDOCK score****C Three best HADDOCK models**
Fig. S10. HADDOCK models of BRICHOS-A β 42 complexes. (A) AlphaFold2 model of R221E Bri2 BRICHOS aligned with the Alpha2 model of full-length Bri2 protein where the natural amyloidogenic client, the Bri23 peptide, is shown. Residues close (within 5 Å) to Bri23 peptide are marked in red. **(B)** Haddock score of modeled complexes is presented. **(C)** The three best Haddock models are visualized where the interacting residues on A β 42 fibril are marked in orange and on BRICHOS in red. The zoom-in graph shows the ionic interaction network between K28 on the A β 42 fibril surface and D139, D141 and D158 of BRICHOS.

Reviewer #3 (Remarks to the Author):

The search for means to combat protein assembly diseases was immeasurably advanced with the discovery of a secondary nucleation pathway, which had the two-fold effects of accelerating the process and accounting for the bulk of the fibril mass. The present work takes this mechanism to a new level by the identification of structural origins of the secondary nucleation process, thanks to the adherence of BRICHOS, a domain found in 13 protein families with chaperone activity. Though BRICHOS suppression of secondary nucleation has previously been demonstrated, here solid-state magic angle spinning NMR demonstrates that affected residues are close and located in the three C-terminal beta strands. Model building and paramagnetic resonance enhancement experiments supported these findings. This is thus a well documented and very exciting finding that can have an impact in developing therapeutics as well as probing the fundamental features underlying this assembly.

The only caveat here is that fibrils were created by agitation, and there remains a remote chance that the agitation favors a pathway that, while real, might not be the central one. Similarly, the electron microscopy entails deposition onto grids that again is likely to disrupt features that are not collinear (e.g. the double fibrils observed). Something like an AFM study on non-agitated solutions could thus be welcome in the future. But this thirst for more data should not be taken as undercutting this elegant piece of work, presented in admirable clarity.

We thank the reviewer for the overall positive assessment and suggestions for future experiments.

While we applied agitation for preparation of the samples reported in the main manuscript, we had previously checked different sample preparations as well (referred to as sample 1 to 3 in SI Table S3), where sample (1) & (2) were obtained without shaking. Remarkably, the ^{13}C - ^{13}C DARR spectra of these samples look basically identical with the agitated sample (3), see SI Fig. S3. Hence, we concluded that agitation did not significantly change the fibril morphology.

We welcome the suggestion about AFM measurements, which could indeed give additional information and be included in a future study.

REVIEWER COMMENTS

Reviewer #1 (Remarks to the Author):

The authors have adequately addressed my previous concerns and substantially improved their manuscript. The additional experiments on Brichos added to preformed amyloid-beta fibrils, and in particular the PRE measurements on these samples, provide much more confidence in the interpretation of the data. The same goes for the additional modelling and analysis of the docking results. Altogether the authors present a plausible model for the binding mode of Brichos to amyloid-beta fibrils, which is likely relevant for secondary nucleation. The suggested salt bridge network around K28 is a particularly interesting finding, increasing the significance of the revised manuscript.

Reviewer #2 (Remarks to the Author):

The authors addressed many of my original questions, and the figures illustrating the BRICHOS– Ab42 complex are now clearer. However, a number of data analyses and descriptions remain superficial, sloppy and potentially inconsistent. First, they did not fully clarify and enlarge the TEM images to allow visual inspection of the fibril widths. In Figure 1D, where is the scale bar for 15 nm? If the short black lines across the two filaments are meant to be the 15 nm scale bars, then the Ab42 only fibril has a much wider diameter than 15 nm. This would contradict the authors' statement of an average width of 13.8 ± 2.9 nm. In Figure 2A, sample (2) fibrils (Ab42 monomers added onto BRICHOS-Ab seeds) have similar widths as sample 1, which is the Ab42 only fibril. This contradicts the authors' conclusion in Figure 1. Please clarify. Third, the authors discuss peak doubling for the BRICHOS-bound Ab42 samples, but Figures S4, S6, S7 do not show assignment of doubled peaks. There is a strange lack of willingness by the authors to properly explain their experimental data to back up their conclusions. Please annotate all residues that exhibit peak doubling in Figure S4B, Figure S6, and Figure S7, using a consistent system (such as I and II) to designate the two forms. Finally, Figure S8 is inconsistent with the caption. The caption claims it shows chemical shift doubling, but the red and green bars are indicated as representing different samples. Where is peak doubling indicated?

Point-to-point response to reviewers' comments

Reviewer #1 (Remarks to the Author):

The authors have adequately addressed my previous concerns and substantially improved their manuscript. The additional experiments on Brichos added to preformed amyloid-beta fibrils, and in particular the PRE measurements on these samples, provide much more confidence in the interpretation of the data. The same goes for the additional modelling and analysis of the docking results. Altogether the authors present a plausible model for the binding mode of Brichos to amyloid-beta fibrils, which is likely relevant for secondary nucleation. The suggested salt bridge network around K28 is a particularly interesting finding, increasing the significance of the revised manuscript.

We thank the reviewer for the positive assessment, and for the reviewer's previous comments, leading to this improved manuscript.

Reviewer #2 (Remarks to the Author):

The authors addressed many of my original questions, and the figures illustrating the BRICHOS– Ab42 complex are now clearer. However, a number of data analyses and descriptions remain superficial, sloppy and potentially inconsistent.

We thank the reviewer for the previous input and the new comments on our manuscript. Based on the new comments we have now updated and improved our manuscript.

First, they did not fully clarify and enlarge the TEM images to allow visual inspection of the fibril widths. In Figure 1D, where is the scale bar for 15 nm? If the short black lines across the two filaments are meant to be the 15 nm scale bars, then the Ab42 only fibril has a much wider diameter than 15 nm. This would contradict the authors' statement of an average width of 13.8 ± 2.9 nm.

We have removed the overview TEM images in Figures 1D and 2A, and substituted them with zoomed images with a scale bar of 50 nm, which allows better visual inspection of the fibrils. Furthermore, we visualized the measurements of the fibril diameters in the zoom-in images of the single fibril. Of note, the measurements of the fibril diameter exhibit a distribution of different fibril diameters. We have now included all measurement values of the dot plots in Figure 1E, Figure 2B and SI Figure S2 as supplementary source data. We noticed that the zoom-in image of the A β 42 alone fibril we showed previously in Figure 1E did not represent an average fibril diameter but was an example of larger diameter within the distribution. We have now replaced that image with a fibril whose diameter represents better the average of the distribution, which should enhance the clarity of the example images, please see updated figures below.

Figure 1. Binding of BRICHOS to A β 42 fibrils. (A) SPR measurements revealing a dissociation constant of BRICHOS to A β 42 fibrils of 12.9 ± 0.2 nM. (B) Solution NMR ^1H - ^{15}N HSQC experiments of ^{15}N -labeled BRICHOS showing an intensity decrease to 90 ± 8 % upon addition of A β 42 fibrils at a 1:1 molar ratio (related to monomeric A β 42). (C) SDS-PAGE analysis of soluble BRICHOS after co-incubation with A β 42 shows that the large proportion of BRICHOS is still soluble. The uncropped SDS-PAGE gel is shown in SI Figure S1. (D) EM images exhibiting thinner fibrils in the BRICHOS-A β 42 sample compared to mature A β 42 fibrils. (E) Fibril diameter showing a reduction of a factor of around two in the presence of BRICHOS. (F) Schematic overview about BRICHOS-modulated A β 42 fibril formation, where BRICHOS predominately inhibits secondary nucleation processes (k_2) in addition to fibril-end elongation (k_+) and favors the generation of thinner fibrils.

Figure 2. Fibril morphology of third generation seeded fibrils. (A) EM images of third generation Aβ42 fibrils prepared using (1) Aβ42 monomers & Aβ42 seeds, (2) Aβ42 monomers & BRICHOS-Aβ42 seeds and (3) Aβ42 monomers & BRICHOS & BRICHOS-Aβ42 seeds. (B) Fibril diameter of third generation seeded fibrils, exhibiting similar diameters of fibrils prepared according to (1) and (2) but half diameter fibrils for preparation (3), indicating that BRICHOS needs to be present during fibril formation to produce thinner fibrils. (C) Schematic overview about BRICHOS-modulated fibril formation using seeding, where mature Aβ42 fibril and BRICHOS-Aβ42 efficiently seed Aβ42 aggregation by promoting secondary nucleation processes (k_2), which is inhibited by the presence of BRICHOS.

In Figure 2A, sample (2) fibrils (Aβ42 monomers added onto BRICHOS-Aβ seeds) have similar widths as sample 1, which is the Aβ42 only fibril. This contradicts the authors' conclusion in Figure 1. Please clarify.

Indeed, the fibrils of sample (2) in Figure 2, produced from fibrillization of Aβ42 monomers using BRICHOS-Aβ42 co-incubated seeds, have a very similar diameter as the non-seeded Aβ42 fibrils shown in Figure 1. Importantly, the BRICHOS-Aβ42 seeds only contain a small fraction of bound BRICHOS (which could be released), and hence the large majority of Aβ monomers aggregate without the presence of BRICHOS, resulting in larger diameter of the fibrils. The interesting finding from this experiment is that the thin fibril structure of the BRICHOS-Aβ42 seeds does not propagate, and to form thin fibrils the presence of BRICHOS in solution is essential, which is represented by sample (3).

Third, the authors discuss peak doubling for the BRICHOS-bound Aβ42 samples, but Figures S4, S6, S7 do not show assignment of doubled peaks. There is a strange lack of willingness by the authors to properly explain their experimental data to back up their conclusions. Please annotate all residues that exhibit peak doubling in Figure S4B, Figure S6, and Figure S7, using a consistent system (such as I and II) to designate the two forms.

We apologize for the unclear labeling. We have previously only assigned/labeled the new peak in the BRICHOS co-incubated Aβ42 fibril sample, where the other peak corresponds to the same assignment as in the Aβ42 fibril alone sample. We have now updated the labeling, introducing a star (*) suffix for the new peaks only visible in the BRICHOS co-incubated Aβ42 fibril sample, in Figure 3 as well as

for SI Figures S6 and S7. Due to the broad signals, we did not assign any doubled cross-peaks in ^1H , ^{15}N -correlation spectra (SI Figure S4). We updated the manuscript and the figures accordingly.

Updates in manuscript in ^1H -detected MAS NMR section:

The ^1H , ^{13}C -correlation spectrum of the BRICHOS-A β 42 fibrils exhibits in general broader but still resolved signals and most of the cross-peaks overlay with the signals of A β 42 alone. Yet, peak doubling was observed for several distinct signals (Figure 3A,B), where a first set of peaks overlays with the corresponding resonances in the apo form of A β 42 fibrils, and a second set is shifted.

Updated figures:

Figure 3. Chemical shift changes of BRICHOS-A β 42 fibrils and structural model. (A) ^1H , ^{13}C -correlation spectra of A β 42 fibrils alone (blue) and BRICHOS-A β 42 co-incubated fibrils (red). The doubled cross-peaks in the BRICHOS-A β 42 spectrum are labeled in red with a star (*). The insets represent different zoomed regions. **(B)** Combined (^1H and ^{13}C) chemical shift changes between the new signals and those observable in the spectra of A β 42 fibrils alone. Circles refer to overlap in the spectrum, stars to ambiguous assignments and crosses to residues with missing assignment. **(C)** Residues exhibiting signals with significant chemical shift doubling are colored in orange and yellow (for ambiguous assignments) on the fibril structure ²³, revealing that the last three β -strands, including the salt bridge between K28 and A42, are affected by the presence of BRICHOS. **(D)** Such residues are also illustrated onto the 3D model of tetrameric A β 42 fibrils ²⁴.

Fig. S6. Assignment of shifted peaks of BRICHOS-Aβ42 using (H)CBCAH experiment. Spectra are shown for Aβ42 fibrils alone (blue) and BRICHOS-Aβ42 fibrils (red). Strips of (H)CBCAH experiment of the three doubled peaks, marked with a star, in the CA-region using previous assignment of Aβ42 alone fibrils ¹.

Fig. S7. ^1H , ^{13}C -correlation spectra. The ^1H , ^{13}C -correlation spectra of A β 42 fibrils alone (blue), BRICHOS added to mature A β 42 fibrils (green) and BRICHOS-A β 42 co-incubated samples (red) are shown. The inserts represent different zoomed regions. The assignments of the doubled peaks visible in the BRICHOS samples are marked with a star (*).

Finally, Figure S8 is inconsistent with the caption. The caption claims it shows chemical shift doubling, but the red and green bars are indicated as representing different samples. Where is peak doubling indicated?

The SI Figure S8 shows the chemical shift differences of the new peaks (labeled with a star) and the original peaks. These chemical shift differences are shown for two different samples, represented as red and green bars. We have updated the figure legend to clarify this issue.

Fig. S8. Chemical shift differences between doubled peaks in ¹H,¹³C-correlation spectra. Combined (¹H and ¹³C) chemical shift changes between the set of signals of mature Aβ42 fibrils alone and the new set of peaks visible in co-incubated BRICHOS-Aβ42 fibrils (red) and those visible in BRICHOS added to mature Aβ42 fibrils (green). Due to the low intensity of the doubled peaks of the BRICHOS added to mature Aβ42 fibrils samples, these peaks were assigned based on the doubled peaks of the co-incubated BRICHOS-Aβ42 fibril sample. Missing assignments are labeled with an asterisk (*), crosses (x) refer to nuclei that are not present in the corresponding residue and overlaps are marked by circles (o) where ambiguous assignments for the residues are stated.